# A subset of type-II collagen-binding antibodies prevents experimental arthritis by inhibiting FCGR3 signaling in neutrophils

Zhongwei Xu [1], Bingze Xu[1], Susanna L. Lundström[2], Àlex Moreno-Giró [1,3], Danxia Zhao[1], Myriam Martin [4], Erik Lönnblom [1], Qixing Li [5], Alexander Krämer[1], Changrong Ge[1], Lei Cheng [1], Bibo Liang[1,5], Dongmei Tong [1], Roma Stawikowska[6], Anna M. Blom [4], Gregg B. Fields [6], Roman A. Zubarev [2] & Rikard Holmdahl [1] ✉

Rheumatoid arthritis (RA) involves several classes of pathogenic auto-antibodies, some of which react with type-II collagen (COL2) in articular cartilage. We previously described a subset of COL2 antibodies targeting the F4 epitope (ERGLKGHRGFT) that could be regulatory. Here, using phage display, we developed recombinant antibodies against this epitope and examined the underlying mechanism of action. One of these antibodies, R69-4, protected against cartilage antibody- and collagen-induced arthritis in mice, but not autoimmune disease models independent of arthritogenic autoantibodies. R69-4 was further shown to cross-react with a large range of proteins within the inflamed synovial fluid, such as the complement protein C1q. Complexed R69-4 inhibited neutrophil FCGR3 signaling, thereby impairing downstream IL-1β secretion and neutrophil self-orchestrated recruitment. Likewise, human isotypes of R69-4 protected against arthritis with comparable efficiency. We conclude that R69-4 abrogates autoantibody-mediated arthritis mainly by hindering FCGR3 signaling, highlighting its potential clinical utility in acute RA.

Rheumatoid arthritis (RA) is a progressive and destructive auto-immune disease that affects up to 0.5% of the population worldwide[1]. Although its prognosis has been substantially improved since the introduction of biological disease-modifying anti-rheumatic drugs (DMARDs) neutralizing TNF-α[2], RA remains to be a life-long disease without an efficient cure despite extensive research.

Currently, the paramount goal of treating RA is to attain early and sustained remission, which is positively associated with superior functional outcomes as well as lower healthcare costs[3,4]. Notwithstanding, non-responders exist regardless of which conventional or biological DMARDs are administered. Moreover, the life-long

administration to suppress the immune system arises safety concerns, especially the incremental risks of serious infections and cancer[5]. Therefore, the demand persists for developing novel therapies targeting earlier or alternative pathogenic pathways.

The development of typical RA is preceded by the production of autoantibodies, mainly anti-citrullinated protein antibodies (ACPA) and rheumatoid factors (RF), but also antibodies against self-proteins within articular cartilage, such as type II collagen (COL2). We previously made a surprising observation that autoantibodies against the citrullinated version (F4-CIT-R) of the COL2 F4 epitope (ERGLKGHRGFT, aa 1126-1136, human: P02458-2, mouse: P28481-3)

[1]Division of Medical Inflammation Research, Department of Medical Biochemistry and Biophysics, Karolinska Institute, Stockholm, Sweden. [2]Division of Physiological Chemistry I, Department of Medical Biochemistry and Biophysics, Karolinska Institute, Stockholm, Sweden. [3]Redoxis AB, Lund, Sweden. [4]Department of Translational Medicine, Lund University, Malmö, Sweden. [5]Center for Medical Immunopharmacology Research, Southern Medical University, Guangzhou, China. [6]Institute for Human Health & Disease Intervention and Department of Chemistry & Biochemistry, Florida Atlantic University, Jupiter, FL, USA. ✉e-mail: rikard.holmdahl@ki.se

were negatively associated with disease activity score-28 (DAS28) in RA patients[6]. Likewise, the mouse antibody against this epitope alleviated arthritis in cartilage antibody-induced arthritis (CAIA)[7], a model that mimics the effector phase of human RA[8].

In the CAIA model, arthritogenic antibodies form immune complexes (ICs) with cartilaginous components in joints, triggering inflammation through Fc gamma receptor (FCGR) signaling[9] and the activation of complement[10]. During the development of arthritis, these deposited ICs are thought to be efficiently cleared by macrophages expressing FCGRs, particularly FCGR2B[11]. Furthermore, the presence of complement component 1q (C1q), which initiates the classical pathway of complement activation, aids in this clearance process[12]. The antibodies targeting the F4 epitope, could potentially affect the development of arthritis if they disturb the functions of these key mediators.

Since the sequence of the F4 epitope is highly conserved in general (identical between humans and mice), we engineered a series of recombinant antibodies against this epitope and tested them in mouse models. Inspiringly, some candidates demonstrated potent abilities to ameliorate CAIA. One of the most effective candidates, R69-4, typically eliminated arthritis within 3 days after injection. Further investigation revealed that R69-4 could complex with various targets, prohibit neutrophil FCGR3 signaling, and break the loop of neutrophil recruitment, which allows for rapid disease remission. Thus, R69-4 possesses a promising therapeutic potential for RA, particularly during its acute phase.

## Results

### Candidate selection

After multiple rounds of screening based on the binding to the triple-helical peptides GFS-5 and GFS-15 that contain the F4 epitope and its citrullinated form, respectively, 265 clones were selected and sequenced, of which 77 clones were unique. The binding to GFS-5, GFS-15 and control peptide has been summarized in Table. S1. Subsequently, these clones plus controls were further screened according to their binding to peptides including synF4-24aa, recF4-24aa, synF4-12aa, as well as to neonatal cartilage tissue (Table. S2). Seven clones including controls were selected for full-length mouse IgG2b (λ) antibody production. These recombinant antibodies were denoted R69, followed by the clone number. Two clones were excluded due to instability or low expression yield, leaving 5 clones finally scheduled for animal experiments (R69-4, −7, −14, −18, and −19). Details of the screening procedures can be found in the Supplementary notes.

### R69-4 protects against CAIA and CIA

To select the most promising candidate, we tested these clones in CAIA model using BQ.Cia9i mice. Of all clones tested, R69-4 provided the best protection against arthritis, whereas R69-18 did not result in significant disease remission (Fig. S1a). Considering the productivity and efficiency, R69-4 was selected as the best candidate for further investigation. The efficacy of R69-4 was thereafter validated separately (Fig. 1a). R69-18 exhibiting a similar binding pattern without protective potential, served as an isotype control.

To determine the best isotype, we class-switched R69-4 to other mouse isotypes. Mouse IgG2b remained the most effective isotype in protecting against CAIA (Fig. S1b). To extend these findings to humans, we class-switched R69-4 to human IgG isotypes, and tested their function in CAIA model, as human IgGs are reported to interact with mouse FCGRs[13]. The human R69-4 isotypes showed varying levels of protection against arthritis, with hIgG2 as the best candidate (Fig. S1c). Its binding to the F4 epitope was validated by bead-based flow

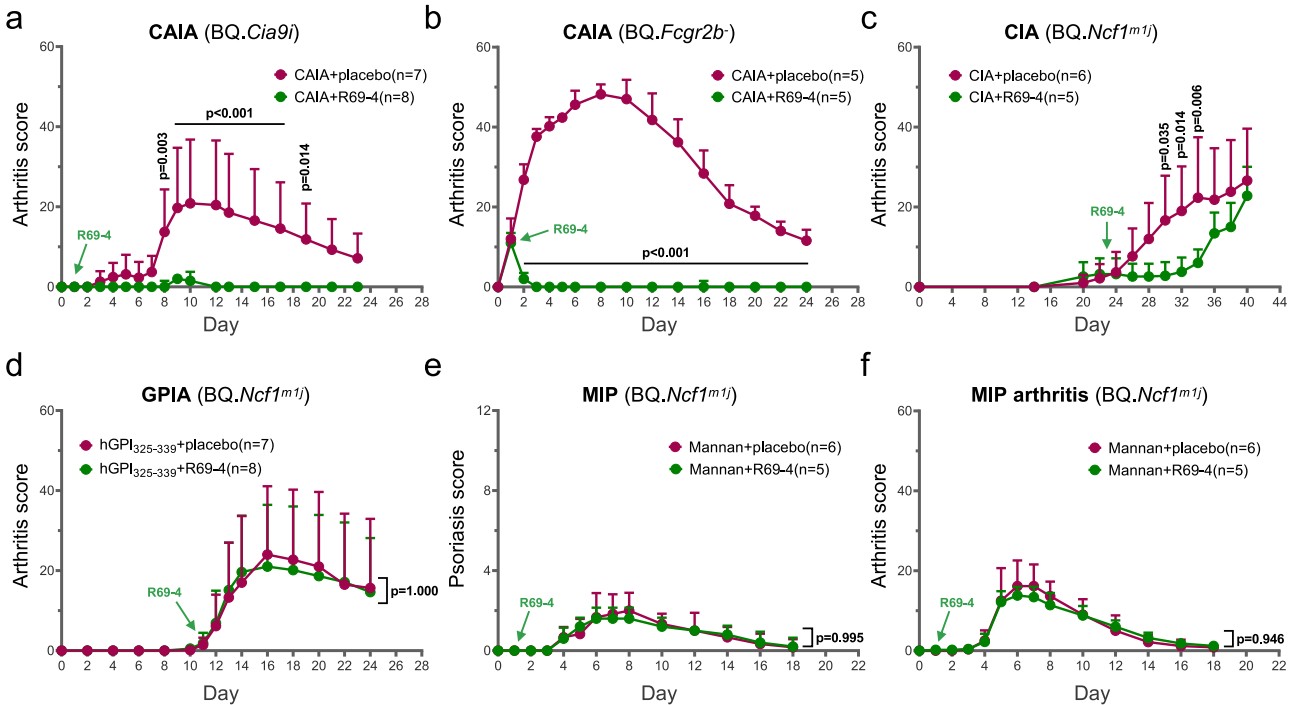

**Fig. 1 | Protective efficacy of R69-4 in multiple mouse models. a** R69-4 protected against cartilage antibody-induced arthritis (CAIA) in BQ.Cia9i mice (Two-way ANOVA); cartilage antibody cocktail (Cab4): 4 mg, d0, i.v.; R69-4: 2 mg, d1, i.v.; lipopolysaccharide (LPS) boost: 25 μg, d5, i.p. **b** R69-4 protected against CAIA in BQ.Fcgr2b⁻ mice (Two-way ANOVA); Cab4: 2 mg, d0, i.v.; R69-4: 1 mg, d1, i.v.. **c** R69-4 protected against collagen-induced arthritis (CIA) in BQ.Ncf1^{m1j} mice (Two-way ANOVA); rat collagen type II (COL2): 100 μg emulsified in complete Freund's adjuvant (CFA), d0, i.d.; boost: rat COL2 (50 μg) emulsified in incomplete Freund's adjuvant (IFA), d21, i.d.; R69-4: 1 mg, d23, i.v.. **d** R69-4 failed to protect against human GPI325-339 peptide-induced arthritis (GPIA) in BQ.Ncf1^{m1j} mice (Two-way ANOVA); hGPI325-339 peptide: 10 μg emulsified in CFA, d0, i.d.; R69-4: 1 mg, d11, i.v.. **e, f** R69-4 failed to protect against mannan-induced psoriasis (MIP) or psoriatic arthritis in BQ.Ncf1^{m1j} mice (Two-way ANOVA); mannan: 20 mg, d0, i.p.; R69-4: 1 mg, d1, i.v.. All plots were shown as mean + standard deviation (SD). Placebo: 0.01 M glycine, 0.15 M NaCl in all plots.

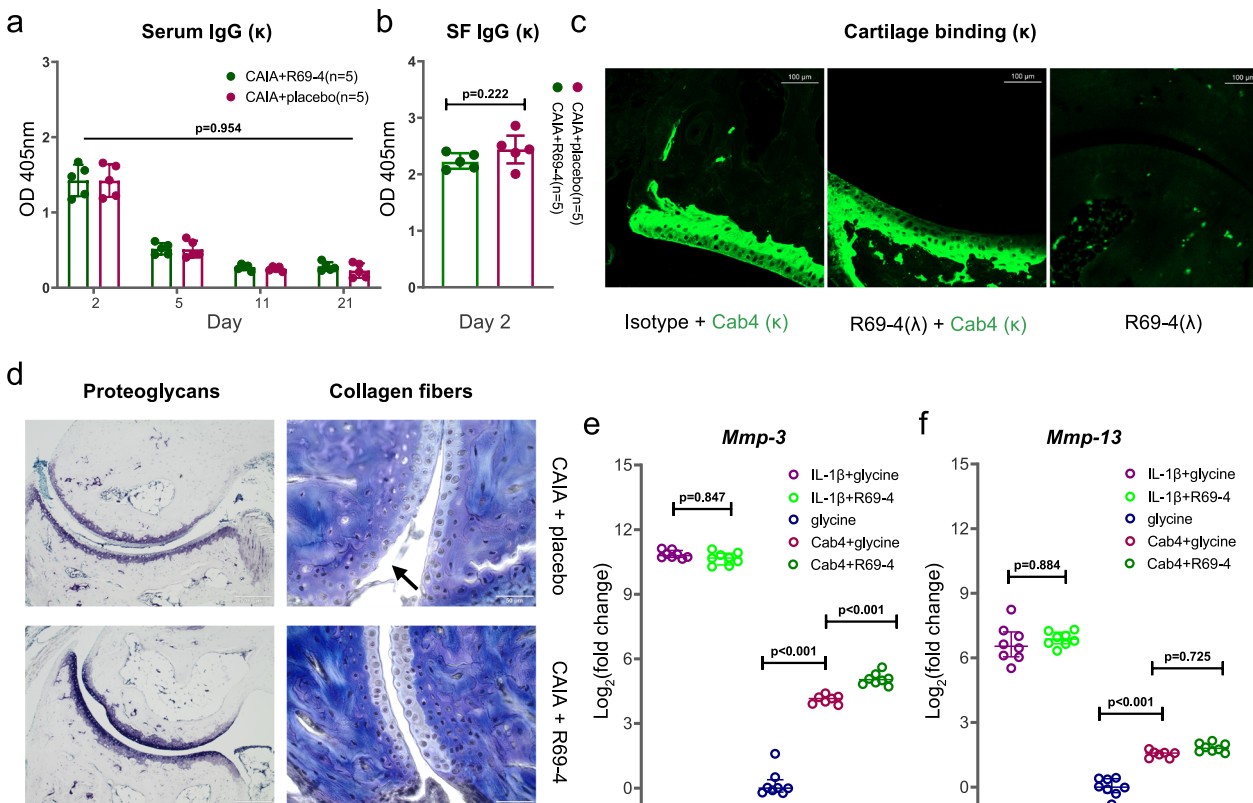

**Fig. 2 | R69-4 protects against cartilage proteoglycan depletion and collagen fiber breakdown, without interfering with pathogenic antibody kinetics and binding. a** The injection of R69-4 did not affect the titers of pathogenic antibodies (Ig κ) in sera at 4 time points, shown as mean ± SD (Two-way ANOVA); Serum samples were diluted by 31,250 times except the ones collected on day 21 (6250 times). **b** Titers of pathogenic antibodies (Ig κ) in synovial fluid (SF) did not change 24 h after R69-4 injection, shown as mean ± SD (Mann–Whitney *U* test, two sided); SF samples were diluted by 625 times. **c** Fluorescent intensity of pathogenic antibodies (IgG κ) binding to cartilage tissue did not differ with or without the pre-incubation of R69-4 (IgG λ); scale bar: 100 μm. **d** R69-4 rescued cartilage breakdown in CAIA; Cab4: 2 mg, d0, *i.v.*; R69-4: 1 mg, d1, *i.v.*; sacrifice: d5; Left: Toluidine blue staining, R69-4 safeguarded against the depletion of cartilage proteoglycans, particularly within the superficial layer; Right: Masson's Trichrome staining, R69-4 protected against cartilage collagen fiber breakdown (arrow); scale bars: 200 μm (left), 50 μm (right); **e** R69-4 elevated the transcription of *Mmp-3* from immature chondrocytes in vitro upon Cab4 stimulation (One-way ANOVA), but not upon IL-1β stimulation (One-way ANOVA); Fold change calculated after transcription levels corrected to β-actin; Data shown as mean ± SD; Cab4: 100 μg/mL, R69-4: 100 μg/mL, IL-1β: 10 ng/mL. **f** R69-4 did not alter the transcription of *Mmp-13* from immature chondrocytes in vitro upon Cab4 stimulation (One-way ANOVA), or upon IL-1β stimulation (One-way ANOVA); Data shown as mean ± SD; Cab4: 100 μg/mL, R69-4: 100 μg/mL, IL-1β: 10 ng/mL.

immunoassay, but with mild cross-reactivity to other epitopes (Fig. S2).

To explore the mechanisms involved in the protective effect, we employed multiple mouse models.

First, we established CAIA in BQ.*Fcgr2b*⁻ mice, which are highly susceptible to arthritis, by the transfer of a cartilage antibody cocktail containing 4 monoclonal antibodies (Cab4). R69-4 injected after onset reversed arthritis development (Fig. 1b), and the injection 24 h before modeling prevented arthritis onset (Fig. S1d). These results confirm its protective effect as well as its independence from FCGR2B. Additionally, the potent protective efficacy against CAIA rules out the involvement of lymphocytes, as CAIA is known to be independent from adaptive immunity[14].

Second, in collagen-induced arthritis (CIA), one dose of R69-4 delayed disease progression (Fig. 1c). However, R69-4 failed to remit inflammation in hGPI₃₂₅₋₃₃₉ peptide-induced arthritis (GPIA) (Fig. 1d), which is known to be driven by T cells[15]. Next, animals were injected with endoglycosidase-S (Endo-S) that can efficiently reduce IgG binding to FCGRs by cleaving Fc glycans[16]. We found that antibody-FCGR interactions do not play a major role in GPIA, as Endo-S injection did not ameliorate arthritis induced by hGPI₃₂₅₋₃₃₉ peptide (Fig. S1e), but nevertheless mitigated inflammation in CAIA (Fig. S1f).

Last, we tested R69-4 in mannan-induced psoriasis (MIP) model, which is driven by activated macrophages and resident γδ-T cells, whereas C5, FCGR3, mast cells, αβ-T cells are all redundant[17]. Likewise, R69-4 failed to protect against psoriasis or psoriatic arthritis (Fig. 1e, f).

Taken together, these observations indicate that the protection provided by R69-4 against arthritis is limited to antibody-mediated arthritis.

## R69-4 does not interfere with pathogenic antibody kinetics or binding

To rule out the possibility of R69-4 facilitating arthritogenic cocktail degradation, we measured the pharmacokinetic profile of the injected pathogenic antibodies. Given that R69-4 protects independently from FCGR2B (Fig. 1b), BQ.*Fcgr2b*⁻ mice were preferably used unless otherwise stated. The serum titers of injected anti-COL2 pathogenic antibodies remained unchanged after the injection of R69-4 throughout the disease periods (Fig. 2a). This suggests that R69-4 did not affect IgG recycling in CAIA, such as through the saturation or blockade of the neonatal Fc receptor (FcRn), which is known to play a crucial role in IgG recycling[18]. Additionally, we compared the pathogenic antibody titers in synovial fluid (SF) 24 h after R69-4 injection, which was not altered either (Fig. 2b), despite R69-4 having already alleviated inflammation.

To determine whether R69-4 competed with the pathogenic cocktail in binding to cartilage, we employed confocal microscopy. The pre-incubation of R69-4 did not reduce the fluorescent intensity of pathogenic antibodies bound to cartilage (Fig. 2c), indicating that R69-4 is unlikely to inhibit the binding from pathogenic antibodies to articular cartilage.

Taken together, these results suggest that R69-4 does not interfere with the kinetics of pathogenic antibodies or their binding to cartilage.

## R69-4 elevates Mmp-3 mRNA transcription

The above findings have shown that after the injection of R69-4 in CAIA, sufficient pathogenic antibodies still circulate and are capable of efficiently binding to cartilage. However, these arthritogenic ICs formed in situ are incompetent to perpetuate inflammation in the presence of R69-4. The downstream cartilage destabilization was, therefore, concerned, which is the first step of antibody-induced pathogenesis[19]. To assess this, we utilized toluidine blue staining and Masson's Trichrome staining to measure cartilage proteoglycan depletion and collagen fiber breakdown, respectively. Consistent with the macroscopic scoring, animals injected with R69-4 after CAIA onset showed better preservation of proteoglycans and collagen fibers (Fig. 2d), indicating that the cartilage breakdown caused by arthritogenic antibodies was halted.

To investigate whether this suspended cartilage breakdown resulted from the inhibition of chondrocyte-derived intrinsic degradation or foreign players from the immune system, we employed an in vitro immature articular chondrocyte (iMAC) system. Here we focused on two relevant enzymes (MMP-3 and MMP-13), due to the vicinity of the MMP-3 binding site to the F4 epitope[20], and the role of MMP-13 in initializing collagen breakdown[21].

After a 48-h culture, chondrocytes incubated with R69-4 transcribed more Mmp-3 mRNA upon Cab4 stimulation (Fig. 2e). R69-4 did not alter the transcriptions of Mmp-13 upon Cab4 or IL-1β stimulation (Fig. 2f). These results suggest that R69-4 plays a typical role as an anti-COL2 antibody, without capacity to lower the secretion of degradative enzymes due to its conformational blockade, in the absence of the immune system.

## R69-4 abrogates antibody induced neutrophil expansion

According to the above observations, the abrogation of inflammation by R69-4 is more likely to result from the suppression of the downstream inflammatory response, rather than the prohibition of chondrocytes derived cartilage degradation. To investigate this, we performed immunophenotyping of peripheral blood mononuclear cells (PBMCs) after the development of arthritis. One day after R69-4 injection, innate cells, particularly neutrophils, were largely reduced in R69-4 injected animals (Fig. 3a). Although restored to some extent, neutrophils from animals treated with R69-4 were constantly fewer until day 4 (Fig. 3b). Moreover, the total neutrophil numbers from both blood and SF were substantially lowered for mice receiving R69-4 treatment (Fig. 3c, d). Since neutrophils play a critical role in CAIA development[22], the suppression of neutrophil expansion can contribute to the rapid disease remission.

## R69-4 prohibits IL-1β secretion in synovial fluid

To investigate how neutrophil expansion was suppressed, we firstly focused on the key mediators for neutrophil chemotaxis in antibody induced inflammation including IL-1β, CXCL2, and CXCR2[23–25].

In CAIA mice, we observed a higher proportion of neutrophils in bone marrow (BM) after R69-4 injection (Fig. S3a), but no difference was recorded regarding their CXCR2 expression, suggesting a reduced demand for BM to release neutrophils[25]. However, chemoattractant CXCL2 was undetectable in serum or SF of either group using cytometric bead array (CBA), arguing against its indispensable role in the

CAIA model. Since the exploration of predominant chemoattractants in CAIA is not the concern of this study, we skipped the screening and measured the upstream key cytokine IL-1β that dictates neutrophil recruitment in autoantibody-induced arthritis[23]. IL-1β was undetectable by CBA in sera of either group, but in SF, abundant IL-1β was detected after CAIA onset. R69-4 treatment strongly inhibited IL-1β secretion (Fig. 3e).

To address whether the substantial reduction of IL-1β release was the primary cause of the protection, we administered recombinant IL-1β to CAIA mice pre-treated with R69-4. IL-1β administration did not restore arthritis suppressed by R69-4 (Fig. 3f), albeit it promoted the migration of immature neutrophils to the periphery (Fig. 3g), indicating that the reduction of IL-1β is secondary to the protective action of R69-4.

To ensure that R69-4 did not undermine neutrophil adhesion to the blood vessel endothelium, which is critical for their migration from periphery to the articular compartment, we measured their CD18 expression. CD18 is required for neutrophil tethering against the shear force of bloodstream[26]. Our results suggest that R69-4 did not affect neutrophil CD18 expression in either circulation or articular compartment (Fig. S3b).

Taken together, R69-4 blocks both IL-1β secretion from joints and neutrophil recruitment to joints, without dampening the capacity of neutrophil migration.

## R69-4 down-regulates FCGR3 expression on myeloid cells

The observed inability of recombinant IL-1β administration in restoring disease suggests that R69-4 may influence the upstream pathways leading to IL-1β production. IL-1β release has been shown to depend exclusively on the engagement of neutrophil FCGR3 (CD16) in autoantibody-mediated arthritis[27]. To investigate whether R69-4 affects FCGR3 signaling, we measured the expression of neutrophil FCGR3 by flow cytometry. One day after R69-4 injection, FCGR3 was largely downregulated particularly on synovial fluid neutrophils (Fig. 4a). Interestingly, it was downregulated in a spatial manner, with neutrophils from the pannus tissue (PT) showing the lowest expression levels (Fig. 4b). Moreover, R69-4 also induced a reduction of FCGR3 expression on infiltrating monocytes, while its effect on macrophages was mild 8 h after R69-4 injection (Fig. S3c).

To exclude a secondary effect of constitutive neutrophil apoptosis causing remarkable FCGR3 downregulation[28] during the resolution of inflammation[29], we measured the apoptotic levels (Annexin-V) of SF neutrophils at an early time point when no disease remission was seen macroscopically. A substantial decrease of FCGR3 was seen on SF neutrophils 8 h after R69-4 injection, however, the majority of the FCGR3[low] neutrophils were Annexin-V negative (Fig. 4c). These results suggest that the R69-4 induced FCGR3 reduction occurred prior to or in the absence of neutrophil apoptosis. Taken together, R69-4 leads to a rapid and substantial FCGR3 reduction on infiltrating myeloid cells, particularly on neutrophils, prior to the resolution of inflammation.

To assess whether the substantial reduction of neutrophil FCGR3 had functional consequences, for example, FCGR3-mediated phagocytosis of ICs[30], we measured the phagocytic capacity of SF neutrophils after R69-4 treatment, using intracellular staining followed by confocal imaging. SF neutrophils isolated from mice treated with R69-4 ingested far fewer IgG2b particles into the cytoplasmic compartment (Fig. 4d), suggesting that R69-4 may hinder neutrophil phagocytosis of ICs in vivo.

## R69-4 depletes neutrophil FCGR3 both in vitro and in vivo

To determine whether R69-4 downregulated FCGR3 by directly interacting with neutrophils, we incubated SF neutrophils with R69-4 in vitro. After a 1-h incubation, a reduction of FCGR3 was recorded in a dose-dependent manner (Fig. 4e). To generalize the finding, we incubated bone marrow derived macrophages (BMDMs) with R69-4

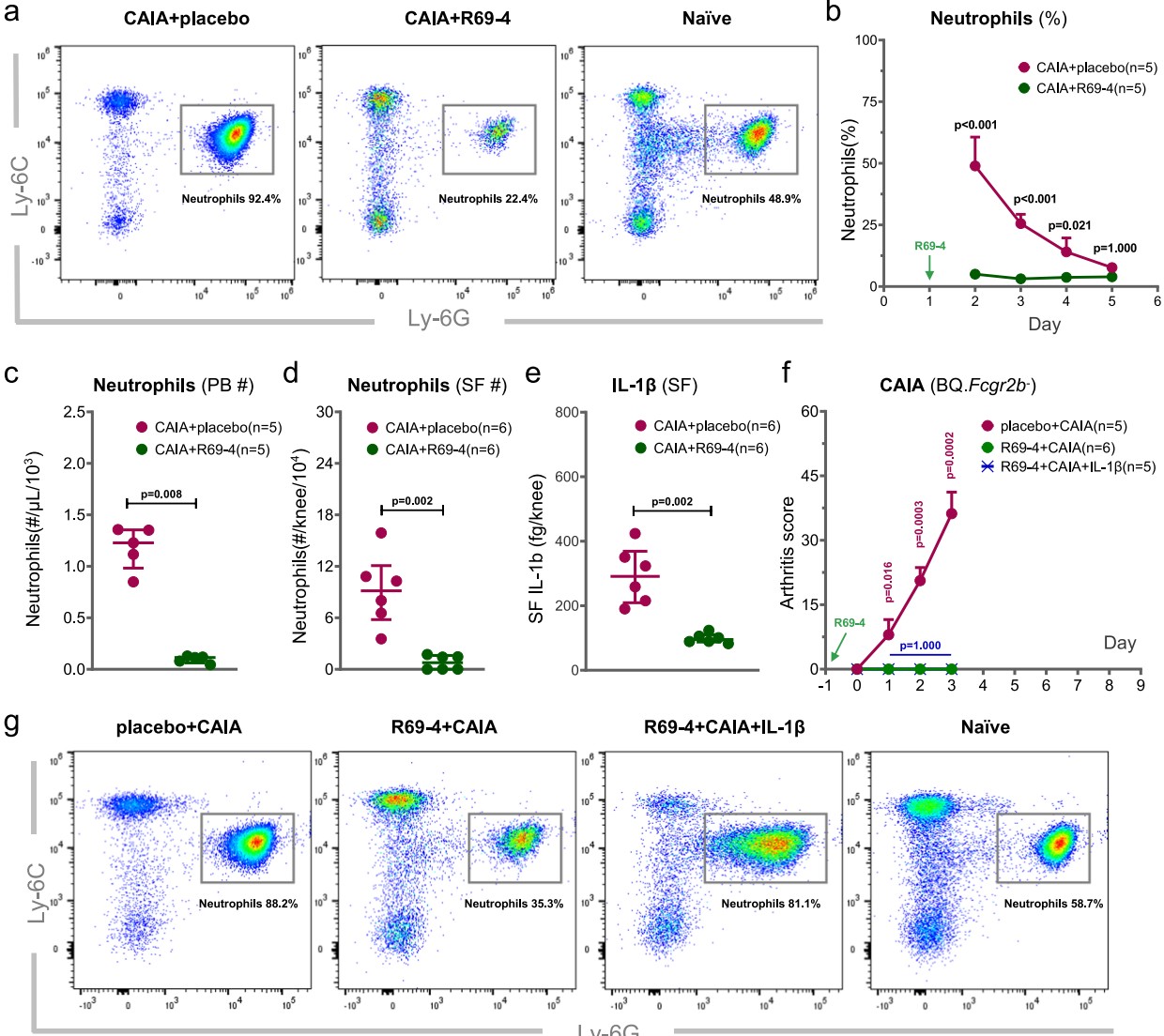

**Fig. 3 | R69-4 abrogates acute granulopoiesis and quenches IL-1β secretion.**
**a** Representative plots showing rapid and potent suppression on neutrophil expansion 24 h after R69-4 injection in CAIA; Events gated from live single CD11b⁺ peripheral blood mononuclear cells (PBMCs); proportions indicate the percentage of neutrophils in CD11b⁺ cells. **b** R69-4 lowered mouse neutrophil proportion of white blood cells (WBCs) in CAIA from d2 to d4 (Two-way ANOVA); Data shown as mean + SD; **c** Mice injected with R69-4 had lower neutrophil absolute counts in PBMCs than that of the untreated CAIA controls (Mann–Whitney U test, two-sided); Data shown as median ± inter-quartile range (IQR). **d** Mice injected with R69-4 had lower neutrophil absolute counts in synovial fluid (SF) than that of the untreated CAIA controls (Mann–Whitney U test, two-sided); Data shown as median ± IQR.

**e** Mice treated with R69-4 demonstrated lower level of IL-1β in SF than that of the untreated CAIA counterparts (Mann–Whitney U test, two-sided); Data shown as median ± IQR. **f** Recombinant IL-1β injection failed to reconstitute disease completely prevented by R69-4; R69-4: 1 mg, d-1, i.v.; Cab4: 2 mg, d0, i.v.; IL-1β: 2.5 µg in PBS containing 0.2% bovine serum albumin (BSA, carrier protein), d1, i.p., same volume of carrier protein was injected to the other two groups; No difference was recorded regarding arthritis severity between the two groups receiving R69-4 (Two-way ANOVA); Data shown as mean + SD. **g** Representative plots showing that the injected recombinant IL-1β was biologically active, which stimulated the release of immature neutrophils into periphery; Events gated from live single CD11b⁺ PBMCs; proportions indicate the percentage of neutrophils in CD11b⁺ population.

in vitro. Macrophages incubated with R69-4 underwent a slower, gradual reduction of FCGR3 (Fig. S4a), indicating that R69-4 has an inferior efficiency in downregulating FCGR3 on macrophages, possibly due to their high plasticity. The effect was also seen on peripheral blood neutrophils isolated from naïve mice without inflammation (Fig. S4b, c).

Thereafter, we followed the dynamic changes of FCGR3 at earlier time points. Surprisingly, neutrophils expressed far higher FCGR3 10 min after the incubation with R69-4; its expression then started decreasing with time (Fig. 4f). In parallel, the intracellular reactive oxygen species (ROS) burst exerted a similar dynamic pattern as FCGR3 expression (Fig. S4d). Intracellular staining revealed upregulated phosphorylation of tyrosine-containing proteins (Fig. S4e),

particularly phospholipase Cγ2 (PLCγ2) upon R69-4 stimulation (Fig. S4f), suggesting a rapid neutrophil activation mediated by immunoreceptor tyrosine-based activation motif (ITAM)[31]. All these observations point to the possible exhaustion of FCGR3 intracellular stock by R69-4, given the fast replenishment of FCGR3 by translocation from the intracellular pools within a short period[32].

To determine whether this effect was valid in vivo, we collected SF neutrophils from mice treated with R69-4. After 10 or 20 min of incubation with R69-4, no increase was seen regarding FCGR3 expression or intracellular ROS burst compared to cells incubated with the isotype control (Fig. 4g and Fig. S5a), pointing to the possible exhaustion of neutrophil FCGR3 by R69-4 in vivo. Taken together, these results show that R69-4 exposure leads to a rapid and

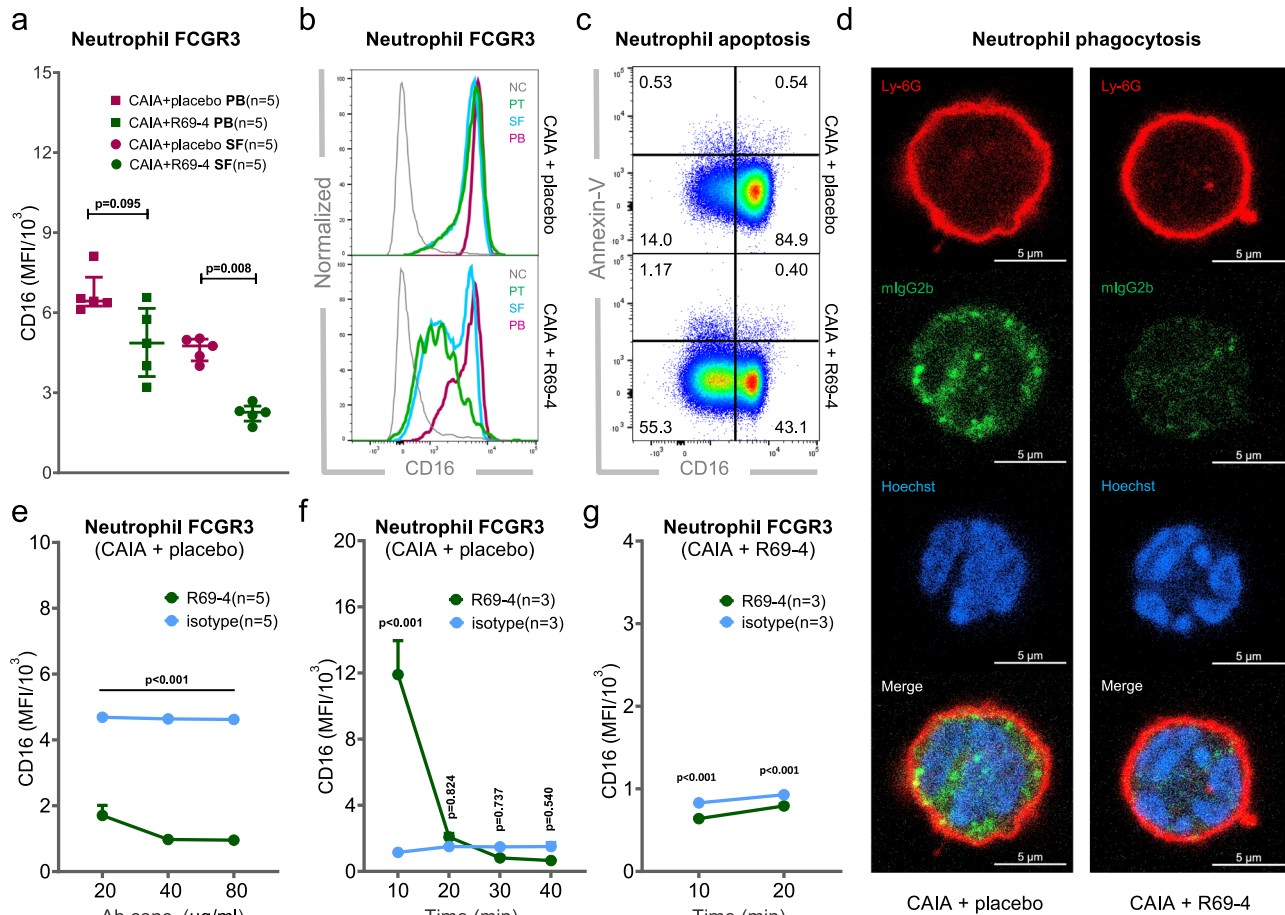

**Fig. 4 | R69-4 reduces neutrophil FCGR3 (CD16) rapidly both in vivo and in vitro. a** R69-4 downregulated FCGR3 expression particularly on SF neutrophils 24 h after injection (Mann–Whitney *U* test, two-sided); Data shown as median ± IQR; **b** FCGR3 expression was downregulated in a spatial manner 8 h after R69-4 injection (pannus tissue (PT) <synovial fluid (SF) <peripheral blood (PB)); negative control (NC): PB lymphocytes; **c** Representative plots indicating that 8 h after R69-4 injection, majority of FCGR3 (CD16)^low neutrophils were not apoptotic (Annexin-V+) in SF; **d** SF neutrophils from mice treated with R69-4 lost competence in phagocytosing IgG2b particles (Green); scale bar: 5 μm. **e** R69-4 reduced SF neutrophil FCGR3 expression efficiently after 1 h of incubation in vitro in a dose-dependent manner, and the lowest dose of R69-4 downregulated neutrophil FCGR3 expression significantly compared to the isotype control (M2139) (Two-way ANOVA); Data shown as mean + SD; **f** SF neutrophils from inflamed CAIA mice exhibited a rapid burst and exhaustion pattern of FCGR3 after incubation with R69-4 (100 μg/ml) but not with the isotype control (M2139) in vitro (Two-way ANOVA); Data shown as mean + SD. **g** SF neutrophils collected from CAIA + R69-4 mice displayed incompetence in responding to R69-4 (100 μg/ml) in vitro (Two-way ANOVA); Data shown as mean + SD. MFI median fluorescent intensity.

substantial reduction of FCGR3 on neutrophils, both in vivo and in vitro.

## R69-4 binds neutrophils through Fc-Fc receptor interaction
To determine whether its protective potential was derived from the direct interaction between R69-4 and FCGR3, we treated R69-4 with Endo-S in vitro to reduce its binding affinity to FCGRs. The enzyme was eliminated by a second-round purification using affinity chromatography. The protective effect of R69-4 was partially impaired after Endo-S hydrolysis (Fig. S5b). To exclude the possibility of potential Endo-S contamination in providing suppression, we mutated an amino acid (position 297) on R69-4 Fc from asparagine (N) to glycine (G), which reduces its relative affinity to all FCGRs by more than 95%[33]. The mutated variant, denoted as R69-4-N297G, showed largely impaired protection against CAIA (Fig. 5a). These results demonstrate the essential role of the interaction between R69-4 Fc portion and FCGRs in providing protection.

IgG antibodies efficiently interact with low-affinity FCGRs only when they form ICs[34]. To investigate how R69-4 interacts with neutrophil FCGR3, we first tested the binding between R69-4 and neutrophils. After a 1-h incubation with R69-4, the fluorescent intensity of IgG λ was elevated on neutrophils (Fig. 5b). This supports the direct

interaction between R69-4 and neutrophils. No clear shift was observed for monocytes incubated with R69-4 (Fig. S5c).

Next, we tested the binding of R69-4 to recombinant FCGR3 by enzyme-linked immunosorbent assay (ELISA). Surprisingly, monomeric R69-4 showed high binding to immobilized FCGR3, whereas the isotype control bound to FCGR3 only in its IC form (Fig. 5c). We then utilized R69-4-Fab and found that the binding to FCGR3 is Fc dependent (Fig. S5d). Taken together, these results suggest that R69-4 can bind neutrophils by direct Fc-FCGR interaction, rather than through specific binding.

## R69-4 bears affinity to diverse potential targets
To determine whether this ligand-receptor binding from R69-4 to neutrophils involved IC formation with proteins within culture media, we cultured neutrophils with R69-4 in protein-free phosphate buffered saline (PBS) only. SF neutrophils from inflamed joints underwent far less ROS burst when incubated with R69-4 in PBS (Fig. 5d), suggesting that R69-4 might be complexed by proteins in culture media prior to interacting with FCGR3 through its Fc portion, which could induce a more potent neutrophil intracellular ROS burst than its monomeric form. To identify the potential cross-reactive targets of R69-4, we employed mass spectrometry.

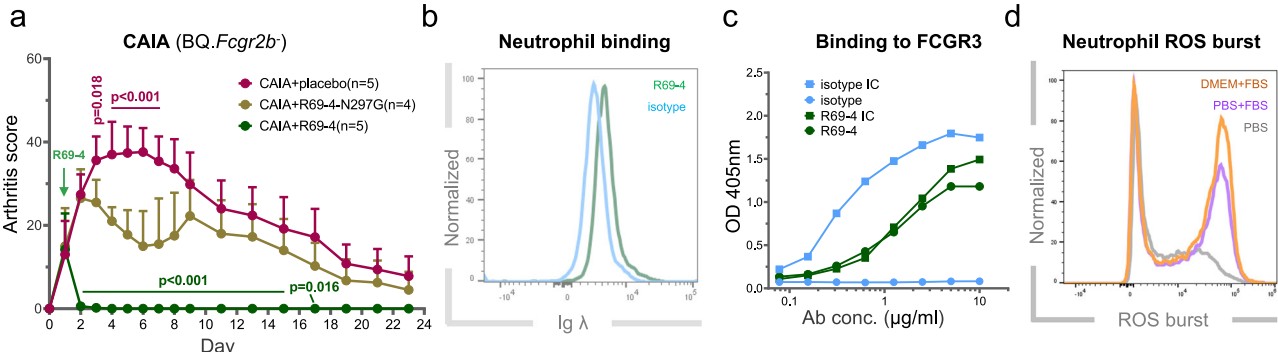

**Fig. 5 | R69-4 Fc-FCGR interaction is essential for its protection. a** Point mutation of R69-4 Fc-N297 to glycine largely impaired its protective efficacy against CAIA in BQ.*Fcgr2b*⁻ mice (Two-way ANOVA, *p* values of post-hoc testing: R69-4 *vs* R69-4-N297G (green), R69-4-N297G *vs* placebo (plum)); Cab4: 2 mg, d0, *i.v.*; R69-4 or R69-4-N297G: 1 mg, d1, *i.v.*; Data shown as mean + SD. **b** Neutrophils incubated with R69-4 for 1 h demonstrated a clear shift on Ig λ fluorescence compared to isotype control (R69-18); **c** Both R69-4 and R69-4 immune complex (IC) bound to FCGR3, whereas the isotype control bound only with its IC form; **d** SF neutrophils underwent lowest reactive oxygen species (ROS) burst when cultured with R69-4 (100 μg/ml, 10 min) in PBS.

Initially, we compared the proteomic profiles of sera and SF supernatants (excluding cells) obtained from CAIA mice. Remarkable decreases in C1q complex components were found in both sera and SF from mice treated with R69-4 (Fig. 6a, b). This finding was further validated by modified CBA in serum (Fig. 6c) and SF samples (Fig. S5e). We then measured the binding between R69-4 and C1q by ELISA. Consistent with the proteomic findings, monomeric R69-4 showed binding to C1q as the control IC (Fig. S5f). R69-4 still bound to C1q when the binding from Fc was prevented by a high-salt solution[35] (Fig. S5g), indicating that R69-4 interacts with C1q through both Fab and Fc parts. These findings suggest that R69-4 is competent in clearing C1q in circulation and the articular compartment, which potentially impedes C1q-mediated opsonization and the classic pathway of complement activation.

To analyze the cross-reactivity more directly, we performed immunoprecipitation (IP) using synovial fluid extracts (including cells) from CAIA mice. More than 500 proteins were precipitated by R69-4 (*p* < 0.05, fold change >1.2) that did not show significance in the isotype control group, indicating its wide-ranging binding capacity to proteins (Fig. 6d). However, most of the precipitated proteins are indiscriminatory between R69-4 and the isotype control (Fig. 6e, *r* = 0.90, 95% CI: 0.89–0.92). Thus, among these proteins many are likely a result of unspecific binding. In contrast, the outliers with fold change (R69-4) > 1.2 and fold change (isotype) <1.2 are then considered as the potential binding targets for R69-4 (Table. S3). FCGR3 is one of these eluted proteins, probably through strong ligand-receptor interaction between FCGR3 and R69-4 which forms ICs with various captured proteins.

To validate these potential targets, we sought to evaluate the binding of all proteins on the list to R69-4. As most of the proposed targets are commercially unavailable, we included alternatives such as fusion proteins and human counterparts. Although R69-4 demonstrated a range of binding to all these (substitute) proteins (Fig. S6a), we advise caution when interpreting the results because of the unavailability of most proteins, coupled with the constraints of IP which may not represent an actual in vivo situation.

The IP results confirm the binding of R69-4 to C1q subunits. Combined with findings that C1q knockout has no effect on CAIA development[36], and that the protective effect of R69-4 was not blunted in BQ.*Fcgr2b*⁻*.Hc** mice with C5 deficiency (Fig. 6f), we conclude that R69-4 induced C1q reduction per se is unlikely to be its protective mechanism of action. The formation of R69-4 ICs interfering with FCGR3 signaling is probably the initial effect in providing protection, and the concurrent C1q reduction can contribute to disease remission because of the dampened C1q-mediated opsonization and complement activation.

Additionally, we investigated the binding between R69-4 and various collagen subtypes. We found that R69-4 exhibited binding to all the subtypes we tested, whereas the anti-COL2 isotype control showed exclusive binding to COL2 (Fig. S6b). Histological examinations on multiple tissue types revealed that R69-4 stained not only cartilage, but also bone, lung, and skin tissues (Fig. S6c).

In summary, these results suggest that R69-4 complexed with its wide-ranging targets can efficiently deplete SF neutrophil FCGR3. This depletion leads to the loss of neutrophil competence for self-orchestrated recruitment, contributing to the rapid resolution of inflammation.

## Discussion

In this study, we developed a series of recombinant IgG antibodies against a highly conserved epitope (COL2 F4). Antibody response to the citrullinated F4 epitope (F4-CIT-R) was previously reported to be negatively associated with RA[6]. Among these recombinant antibodies, one of the best candidates, R69-4, demonstrated potent therapeutic capacity in treating arthritis in CAIA mouse model. R69-4 was then illustrated to exhibit a distinct binding pattern to FCGR3, with the potential to exhaust this receptor on neutrophils during acute granulopoiesis, leading to rapid disease remission.

Initially, we hypothesized that the protective mechanism of R69 antibodies involved the conformational blockade of the F4 epitope. However, the hypothesis was thereafter disproved by the observations that **a**. the protection was not correlated with the binding intensity to cartilage tissue; **b**. The non-binder R69-19 offered better protection than certain strong binders; **c**. the best candidate R69-4 increased *Mmp-3* transcription in chondrocytes in vitro; and **d**. the protective efficacy of R69-4 was largely impaired due to an Fc-glycan mutation. As a result, we shifted our focus to the cross-reactivity of R69 antibodies to potential key mediators in the immune system.

Based on the immunophenotyping findings, we traced from the R69-4 induced effect on neutrophils which plummeted in circulation, back to those recruited into the articular compartment, the upstream IL-1β secretion, and the earliest FCGR3 signaling[27,37]. Typically, acute inflammation is characterized by neutrophil infiltration, whereas the remission coincides with neutrophil retreat, which is usually assumed secondary to the resolution of inflammation. However, with R69-4, the effect on neutrophils appears to be primary. Firstly, R69-4 induced a substantial FCGR3 reduction on SF neutrophils just 8 h after injection, preceding neutrophil apoptosis or macroscopic disease remission. Secondly, the dynamics of neutrophils after R69-4 injection in CAIA suggest a preemptive strike followed by a gradual recovery rather than a secondary retreat that tapers off day by day. Lastly, the potent effect

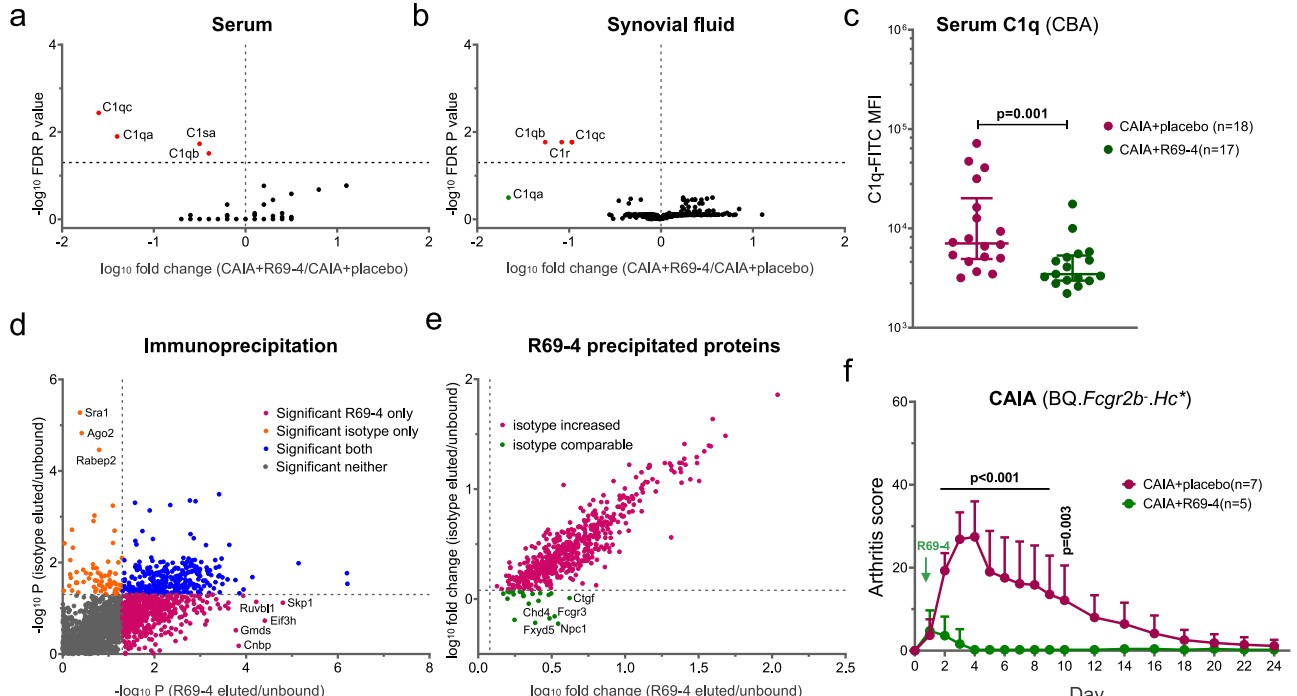

**Fig. 6 | R69-4 bears affinity to diverse endogenous proteins.** Proteomic profiling of sera (**a**) and synovial fluids (**b**) from CAIA mice treated with or without R69-4; Red dots indicate significant decrease after R69-4 injection (Fold change < 1, false discovery rate $p < 0.05$, Student's $t$ test with Storey Tibshirani correction). **c** Modified cytometric bead array (CBA) validation of serum C1q levels; R69-4 injection reduced serum C1q levels in CAIA mice (Mann–Whitney $U$ test, two-sided); Data shown as median ± IQR; **d** Mass spectrometry screening of proteins precipitated by R69-4 or isotype control (R69-18); Plum dots indicate a significance in R69-4 group ($p < 0.05$, Student's $t$ test) but not in isotype control group ($p > 0.05$, Student's $t$ test). **e** Correlation analysis between fold changes of proteins eluted by R69-4 or isotype control ($r = 0.90$, 95% CI: 0.89−0.92); Green dots indicate potential binding targets of R69-4. **f** R69-4 protected against CAIA in BQ.*Fcgr2b⁻.Hc\** mice (Two-way ANOVA); cartilage antibody cocktail (Cab4): 4 mg, d0, *i.v.*; R69-4: 2 mg, d1, *i.v.*; Data shown as mean + SD.

on FCGR3 reduction was observed in direct in vitro cell culture with neutrophils, and this was validated on neutrophils from naïve animals without arthritis, where no secondary effect could be attributed to the resolution of inflammation.

In this scenario, the primary prohibition on neutrophil FCGR3 signaling is believed to contribute to the rapid disease remission[27]. It has been shown that the deletion of FCGR3 results in largely ameliorated inflammation observed in autoantibody induced arthritis[9]. The critical role of FCGR3 in arthritis development is also demonstrated through BM reconstitution, where acute arthritis is not restored by the transfer of FCGR3 deficient BM cells in mice after sublethal irradiation, but rapidly restored after the transfer of FCGR3 sufficient cells[27]. Additionally, we blocked FCGR3 in vitro using a specific antagonist and observed that synovial fluid neutrophils underwent far less ROS burst when incubated with R69-4, suggesting a critical role of FCGR3 in R69-4 induced effects. Here the causality (R69-4 → FCGR3 ↓ → arthritis ↓) is firmly ascertained. Since FCGR3⁻ mice failed to develop CAIA, we were unable to determine whether this pathway is exclusively responsible for the observed protection. Therefore, other potential mechanisms may also exist, for example, the suppression of FCGR4. This receptor is reported to contribute especially to IgG2b-mediated inflammatory responses[38], and the deletion of FCGR4 reduces neutrophil recruitment in immune-complex-mediated peritonitis[39]. We observed that R69-4, with IgG1 isotype that does not interact with FCGR4[13], still protected against arthritis, which excludes the necessity of its interaction with FCGR4 in providing protection. However, R69-4 with IgG2b isotype demonstrated superior efficiency in suppressing CAIA compared to its IgG1 counterpart, suggesting that its interaction with FCGR4 may contribute to the protective effect to some extent.

It is important to note that the effectiveness of R69-4 was not diminished even in the absence of C5. Complement C5 is a crucial component that undergoes cleavage into C5a and C5b during complement activation. While C5b primarily participates in the terminal pathway of the complement activation cascade, C5a plays a more significant role in the lipid-cytokine-chemokine cascade, contributing to neutrophil recruitment in antibody-mediated inflammation[27]. Mice deficient with C5 experience difficulties in releasing either C5a or C5b. Therefore, we speculate that neither the C5a-C5aR-LTB4 axis nor the formation of the membrane attack complex (MAC) is involved in the protective effect of R69-4 during the development of CAIA.

The slight neutrophil ROS burst induced by monomeric R69-4 in PBS raises concerns on potential IgG aggregates, since this effect resembles that of insoluble ICs in vitro[40]. To address this concern, we subjected R69-4 to high-speed centrifugation to remove any potential aggregates[40], and the resulting "cleaned" monomers still protected against arthritis. These monomers in vitro downregulated FCGR3 efficiently (Fig. S4b). Interestingly, when we pre-aggregated R69-4 in vitro, we found that the fully aggregated suspension without soluble monomers still conferred protection against CAIA. These observations offer a promising perspective on the role of IgG aggregates in regulating inflammatory diseases.

To translate these findings into clinical applications, we isotype-switched R69-4 to human IgGs and observed equivalent protection against CAIA. This may be attributable to the high affinity between mouse FCGRs and human IgG isotypes[13], which is comparable to the binding between mouse FCGRs and mouse IgGs. Following R69-4 administration, we observed a potent suppression on mouse FCGR3. Its human ortholog, FCGR2A[41], having been demonstrated to govern TNF-α production in response to IgG ICs[42], may have a role in RA.

The acute phase of RA is a critical period to seek prompt medical attention, as early and efficient intervention can slow the progression of the disease and improve long-term prognosis[43]. However, inducing sustained remission can be challenging in some RA patients. Nearly half of naïve RA patients with early active disease respond poorly to the TNF inhibitor infliximab in combination with anchor drug methotrexate (MTX)[44]. R69-4 might be a viable option to compensate for this unsatisfactory efficiency. During RA onset, aggressive neutrophil infiltration largely contributes to synovitis and joint damage. Depleting neutrophils is nevertheless impractical as well as dangerous to the host's defense. Instead, R69-4 that hinders deposited ICs from interacting with neutrophil FCGRs, will be an outstanding alternative to prevent the perpetuation of neutrophil recruitment, thereby discontinuing disease progression. Likewise, R69-4 may also protect against RA flares.

However, autoantibodies are more likely to drive inflammation in acute relapsing RA rather than during its chronic progressive phase. B cell depletion using anti-CD20 monoclonal antibody is not universally effective in RA patients[45]. In contrast, it is suggested that other mechanisms, mediated by T cells, macrophages or fibroblasts, may play a major role in the chronic phase of arthritis[46,47]. Interestingly, in the T cell-mediated or macrophage-mediated arthritis models, R69-4 has been demonstrated to be ineffective, rendering it insufficient as a monotherapy for chronic arthritis. Instead, using R69-4 in acute arthritis or combining it with existing therapies may improve the long-term management of RA, offering better efficacy in the treatment approach.

Thus, we propose that antibodies to the F4 epitope on COL2, cross-reactive with various proteins in joints, with the ability to prohibit neutrophil FCGR3 signaling, represent a new therapeutic opportunity for RA, especially during its acute phase.

## Methods

### Antibody production

Mouse arthritogenic antibodies against COL2 or cartilage oligomeric matrix protein (COMP) were provided by Vacara AB (Stockholm, Sweden) including M2139 (mIgG2b), ACC1 (mIgG2a), 15 A (mIgG1), and L10D9 (mIgG2b). Rat anti-mouse CD16/32 antibody (rIgG2b) was produced in house by the corresponding hybridoma (2.4G2).

Recombinant antibodies against the F4 epitope were constructed using single-chain variable fragments (scFv) sequences selected from two phage display libraries. Briefly, peptides containing the triple-helical F4 epitope[48] were used for selection, including GFS-5 and GFS-15. The GFS-5 peptide is flanked by 5 GPO repeats on both N- and C-terminus. The C-terminus is biotinylated and at the C-terminus of the 5 GPO repeats, a lysine knot is added to covalently link the peptide to form a stable triple-helical structure. The GFS-15 has the same arrangement as GFS-5 except the first arginine residue that is replaced by citrulline. GFS-2 peptide containing a triple-helical C1 epitope was used as a counter screen. To this end, scFv phage display library technology was used to select clones that specifically target the F4 epitope. After selection, the binding was validated using surface plasmon resonance (SPR), bead-based flow immunoassay, and immunohistochemistry (IHC). The selected scFv candidates were converted to full-length antibodies by inserting the variable region of heavy chains to vectors containing a mouse IgG2b constant region with KpnI/HindIII restriction sites, and the variable region of light chains to vectors containing a mouse lambda (λ) constant region with KpnI/NheI sites, respectively. The constructs were then transformed into DH5α chemical competent cells for plasmid amplification, and large-scale production of full-length antibodies was carried out by transfecting Expi293F cell line (ThermoFisher Scientific) using FectoPro transfection reagent (Polyplus Illkirch, France) with corresponding plasmids. Recombinant antibodies in the cell culture supernatants were harvested 5 days after transfection, purified using GammaBind Plus

Sepharose antibody purification resin (Cytiva), followed by dialysis against 0.01 M glycine solution containing 0.15 M NaCl to avoid aggregates, and thereafter concentrated, buffered in the same glycine solution, and finally stored at 4 °C before injection. Once converted to full-length IgGs, the recombinant antibodies were denoted R69-clone number. The detailed screening procedure can be found in the Supplementary notes.

The in vitro Fc-glycan hydrolysis was carried out by incubating PBS buffered R69-4 (1 mg/mL) with Endo-S (2 µg/mL) for 1 h at 37 °C. The enzyme was then removed by passing the mixture through a Protein-G column linked to an ÄKTA chromatography system. The purified antibody was then dialyzed as described above and concentrated before injection.

### Animal models

Mice with a deleted *Fcgr2b* gene[49] or a mutated *Ncf1* gene (*Ncf1^mlj*)[50] on the C57BL/6 J (B6) background were obtained from Jackson Laboratory, and then backcrossed into C57 black mice with H2^q derived from DBA/1 (abbreviated BQ) by more than 10 generations (BQ.*Fcgr2b*⁻ or BQ.*Ncf1^mlj*). BQ.*Cia9i* mice were generated by inserting the *Cia9* congenic fragment with an FcR haplotype of *Mus musculus* origin into BQ mice, as described previously[51]. Mice deficient for both FCGR2B and complement C5 (BQ.*Fcgr2b*⁻.*Hc*ˣ) were established by crossing *Fcgr2b* knockout and *Hc* mutated mice[52]. All animals included in this study were healthy and active males aged 10–16 weeks, and those reaching any humane endpoints during experiments were sacrificed. Sample sizes were determined according to the 3 R guidelines (Replacement, Refinement, and Reduction). Animals were grouped by generating random numbers in each cage, and littermate control strategy was applied to minimize potential confounders, following the ARRIVE protocol[53].

All animals were housed in FELASA2 specific pathogen-free (SPF) facility in the Comparative Medicine's Annex (KM-A) at Karolinska Institute, under the following conditions: light/dark cycle (dawn 6:00–7:00, day 7:00–18:00, dusk 18:00–19:00, night 19:00–06:00), relative humidity 50%, and ambient temperature 22 °C. All experiments were carried out according to the ethical permits (No: N35/16 and 2660-2021) approved by the animal ethics committee (Stockholm region) affiliated with the Swedish Board of Agriculture.

CAIA models were established in BQ.*Cia9i*, BQ.*Fcgr2b*⁻.*Hc*ˣ, or BQ.*Fcgr2b*⁻ mice. For CAIA induction (day 0), 4 mg of antibody cocktail containing M2139, ACC1, 15 A, and L10D9 (denoted Cab4) was administrated intravenously (*i.v.*) into BQ.*Cia9i* mice, and 25 µg of lipopolysaccharide (LPS, Sigma-Aldrich) was injected intraperitoneally (*i.p.*) on day 5 for boost. Recombinant antibody R69-4 (2 mg/100 µL) or same volume of placebo (0.01 M glycine, 0.15 M NaCl) was injected *i.v.* on day 1. The placebo used in all other models was consistent unless otherwise noted. The procedures for BQ.*Fcgr2b*⁻.*Hc*ˣ mice were same as above but without LPS boost. Regarding BQ.*Fcgr2b*⁻ mice, halved dose of Cab4 (2 mg) was administrated *i.v.* on day 0, and halved dose of recombinant R69-4 antibody (1 mg, 100 µL) or same volume of placebo was administrated *i.v.* on day 1. Recombinant IL-1β (2.5 µg, Biolegend) in PBS containing 0.2% BSA (carrier protein) was injected *i.p.* in one CAIA experiment, and the control mice received same amount of the solvent with carrier protein only. LPS boost was not administrated to BQ.*Fcgr2b*⁻ mice considering animal welfare issues (high disease severity). Arthritis score was assessed blindly according to a 60-point scale[54] every 1 to 2 days.

CIA models were established in BQ.*Ncf1^mlj* mice. Briefly, 100 µg (diluted in 50 µL PBS) of rat COL2 (in-house production) emulsified in 50 µL of complete Freund's adjuvant (CFA, Difco, Nordic BioLabs) was injected intradermally (*i.d.*) at the base of the tail (left) for all mice on day 0, and on day 21, 50 µg (50 µL) of rat COL2 emulsified in 50 µL of incomplete Freund's adjuvant (IFA, Difco, Nordic BioLabs) was injected *i.d.* at the base of the tail (right) for boost. Two days after the boost,

recombinant antibody R69-4 (1 mg) or placebo was injected *i.v.*, and the arthritis severity was scored blindly every 2 days.

GPIA models were also established in BQ.*Ncf1^{m1j}* mice. Like CIA, 10 μg (diluted in 50 μL of PBS) of hG6PI$_{325-339}$ peptide (-IWYINCFG-CETHAML-)[55] emulsified in 50 μL of CFA was injected *i.d.* into the base of tail on day 0, but no boost was carried out. R69-4 (1 mg) or placebo was injected *i.v.* on day 11 right after the disease onset. Arthritis severity was scored blindly every 1 to 2 days.

MIP models were established in BQ.*Ncf1^{m1j}* mice by the injection of 20 mg of mannan (Sigma-Aldrich, dissolved in 200 μL of PBS, *i.p.*) on day 0. R69-4 (1 mg) or placebo solution was injected *i.v.* on day 1. Psoriasis and psoriatic arthritis severity were scored blindly every 1 to 2 days according to a 12-point and 60-point scale, respectively[56].

### ELISA

The titers of serum or SF pathogenic antibodies were determined by ELISA. Rat COL2 diluted in PBS (5 μg/mL, 100 μL) was coated into each well of 96-well MaxiSorp NUNC ELISA plates (Thermo Scientific) and incubated overnight at 4 °C, and the wells were then blocked by 3% milk powder (Merck) dissolved in PBS for 1 h at ambient temperature. Subsequently, 100ul of diluted serum samples (1:6,250 or 1:31,250 in PBS) or SF samples (1:625 in PBS) were applied to the wells and incubated for 1 h at 37 °C. Rat anti-mouse kappa (κ) antibody conjugated with horseradish peroxidase (HRP) (1:2,000 in 3% milk, Southern Biotech) was used to detect the bound COL2 specific antibodies (100 μL, 1 h at 37 °C). ABTS substrate (50 μL, 1 mg/mL in ABTS buffer, Roche, Merek) was added to visualize HRP conjugated to the secondary antibody. Four times of wash were applied using PBS + 0.05% Tween (Sigma-Aldrich) after each step above. After a 5-min incubation of ABTS substrate at ambient temperature, the optical density (OD 405 nm) values of all wells were measured using Biotek Synergy 2 microplate reader.

The binding of R69-4 to mouse FCGR3/CD16 was measured by ELISA. His-tagged recombinant mouse FCGR3 (100 μL, 5 μg/mL, BioLegend, Thermo Fisher) was added into wells of 96-well Ni-NTA His-Sorb ELISA plates (Qiagen) and incubated overnight at 4 °C. The plate was then washed and blocked by 3% milk for 1 h at ambient temperature. R69-4, R69-4-IC (R69-4-COL2), isotype control (M2139), or isotype-IC (M2139-COL2) (100 μl, 10 μg/mL) were added into wells, followed by gradient dilution. After a 1-h incubation of primary antibodies at 37 °C, 100-μL of rat anti-mouse IgG2b secondary antibody conjugated with HRP (1:2,000 in 3% milk, Southern Biotech) was added to the wells and incubated for 1 h at 37 °C. Secondary goat anti-mouse IgG λ conjugated with HRP (Southern Biotech) was used for the detection of R69-4-Fab. Finally, ABTS substrate, washing procedures, and Synergy 2 reading were performed as previously described.

The binding of R69-4 to C1q was measured by ELISA in a similar way, with the additional use of 1 M NaCl solution to prevent the binding between C1q and IgG Fc[35]. Purified mouse C1q (100 μL, 5 μg/mL, Comp Tech) was used for coating, and the following steps were the same as previously described.

The validation of the potential targets of R69-4 was carried out using ELISA with the same protocol. The proteins used for coating include: CORO1C fusion protein (Ag6546, Proteintech), recombinant CHD4 protein (OPPA00068, Aviva Systems Biology), recombinant mouse CTGF (PFT-78NKYJ, Nordic Biosite), recombinant human FXYD5 (PAT-JJG6QK, Nordic Biosite), and Cathepsin B (CTSB) which was made in house using recombinant technology. The concentrations of proteins for coating plates were based on the manufacturers' instructions.

### Cell culture and real-time quantitative polymerase chain reaction (qPCR)

Immature articular chondrocytes (iMAC) were isolated by digesting tibial plateau cartilage tissue pieces from *Fcgr2b^-* neonatal mice aged 5-6 days, using high-concentration Collagenase D (3 mg/mL, Sigma Aldrich) in DMEM media plus antibiotics for 45 min at 37 °C with 5% $CO_2$, followed by a low-concentration Collagenase D (0.5 mg/mL) digestion overnight under the same condition. The details are described previously[57]. Chondrocytes were then resuspended and seeded into 48-well sterile plates (Falcon, AH Diagnostics), and cultured in DMEM media containing antibiotics and 10% fetal bovine serum (FBS) at 37 °C. Tested antibodies or control solvents (0.01 M glycine, 0.15 M NaCl) were added into the cell culture on day 5 when the chondrocytes reached confluence. After 48 h of incubation, chondrocytes were lysed, and total RNA was extracted using RNeasy Plus Kit (Qiagen) according to the manufacturer's instructions. The total RNA was then reverse transcribed to cDNA using the iScript cDNA Synthesis Kit (Bio-Rad) according to the manual. The relative quantification of *Mmp-3* and *Mmp-13* cDNA was performed based on Bio-Rad CFX96 real-time system with iQ SYBR Green Supermix (Bio-Rad) as the fluorescent dye. Relative transcriptions of *Mmp-3* and *Mmp-13* were corrected to β-actin.

Primers used in qPCR were *Actb* (Forward: 5′-CATTGCTGA-CAGGATGCAGAAGG-3′, Reverse: 5′-TGCTGGAAGGTGGACAGTGAGG-3′), *Mmp-3* (Forward: 5′-TTTAAAGGAAATCAGTTCTGGGCTATA-3′, Reverse: 5′-CGATCTTCTTCACGGTTGCA-3′), *Mmp-13* (Forward: 5′-TG ATGGCACTGCTGACATCAT-3′, Reverse: 5′-TGTAGCCTTTGGAACTGC TT-3′).

Macrophages were differentiated from bone marrow cells collected from femurs of adult BQ.*Fcgr2b^-* male mice. Briefly, bone marrow cells were flushed out from femoral medullary cavity using culture media (DMEM + 10% FBS), and tissues were removed by passing through cell strainers with 70-μm pores (Corning, Sigma Aldrich). The single cell suspension was then seeded in 96-well plates and cultivated in the same culture media plus macrophage colony-stimulating factor (M-CSF) (20 ng/mL, PeproTech, ThermoFisher Scientific) for 5 days. The media containing the growth factor were refreshed every 2 days. Bone marrow derived macrophages (BMDMs) were then incubated with R69-4 (100 μg/mL) or isotype control (E0320E7, IgG λ, BioLegend). After incubation, macrophages were detached from plate using 0.05% Trypsin-EDTA (ThermoFisher Scientific) and stained using master mix containing APC anti-mouse F4/80 (BM8, BioLegend).

Neutrophils were collected from PB, SF, or BM of adult mice. ACK buffer was used to lyse red blood cells (RBCs) of PB samples to prepare single-cell suspension. Cell strainers with 70-μm pores (Corning, Sigma Aldrich) were used to filter cells from SF or BM. Cells were resuspended in DMEM media containing 10% FBS plus antibiotics, seeded into 96-well sterile plates (Falcon, AH Diagnostics), and cultured at 37 °C with 5% $CO_2$. Antibody concentration and incubation time were described in detail in corresponding experiments.

### Mass spectrometry

**Sample collection.** Mouse serum samples were collected by centrifugating full blood at $8000 \times g$ for 30 min. SF samples were harvested from bilateral knee joints of CAIA mice and buffered in PBS, and then cells were separated from fluids by centrifugation at $400 \times g$ for 5 min. In terms of immunoprecipitation, concentrated protein extracts were collected by lysing SF samples (containing cells) using Pierce RIPA buffer (Thermo Scientific) for 5 min on ice. Protein-G super-paramagnetic beads (Invitrogen, ThermoFisher Scientific) were pre-saturated by R69-4 or isotype (R69-18) and then incubated with the protein extracts for 10 min. The beads and supernatants (unbound molecules) were separated using a magnetic rack. Proteins bound to R69-4 beads were finally eluted with 1 M acetic acid after adequate wash. All samples were kept under −80 °C prior to proteomic analysis.

**Proteomics sample preparation.** Samples (protein elutes, unbound proteins and pooled SF sample from all mice, prior protein IP-enrichment) were reduced in 8 mM dithiothreitol (55 °C, 45 min) and

alkylated in 25 mM iodoacetamide (25 °C, 30 min in darkness). Protein precipitation was then performed using sample:cold acetone in a volume ratio of 1:6 at −20 °C overnight. The precipitates were collected and washed with acetone by centrifugation (10 min, 10,000 g). Each dry pellet was then resuspended in 15 µL of 8 M urea in 20 mM 4-(2-hydroxyethyl)-1-piperazinepropanesulfonic acid (EPPS) buffer. To the protein elutes, 500 mL of 20 mM EPPS buffer (pH 8.2) was then added, and the samples underwent two additional buffer exchange filtration steps (14,000 g, RT, 30 min, Amicon Ultra 0.5 centrifugal filters, 3 kDa) to remove the Tris buffer. To the unbound proteins and pooled SF sample 15 µL of 20 mM EPPS buffer containing 0.67 µg LysC enzyme (Wako, USA) was added (pH 8.2). The protein elutes were adjusted to contain half the final concentration of LysC but in the same sample volume (30 µL) as the other samples. The digestion with LysC was performed at 30 °C for 8 h. 90 µL of EPPS buffer containing 0.5 µg (protein elutes) or 1 µg (unbound proteins and pooled SF sample) of sequencing grade modified trypsin (Promega, USA) was then added to the samples. The samples were then digested overnight at 37 °C. Approximately 2/3 of each sample was then labeled using two TMT16 (Thermo Fisher Scientific) labeling sets with 16 channels each. Each set contained one blank channel, an aliquot of the pooled SF sample (which was used as a linker to adjust for variation between the two TMT sets) and then half of the protein elutes and half of the unbound proteins, respectively. Sample desalting and cleaning was performed using C18 (Sep-Pak, C18, Vac 1 cm$^3$, 50 mg, Waters, USA). High pH reversed phase fractionation was then performed to separate the peptides using a Dionex Ultimate 3000 system (Thermo Scientific, Germany) on a Xbridge Peptide BEH C18 column (length, 25 cm; inner diameter, 2.1 mm; particle size, 3.5 µm; pore size, 300 Å; Waters) with a flow rate of 200 µL/min. Fractionation was applied using a binary solvent system consisting of 20 mM $NH_4OH$ in $H_2O$ (solvent A) and 20 mM $NH_4OH$ in acetonitrile (solvent B). Proteins were eluted with a gradient from 2% to 23% B in 42 min, to 52% B in 4 min, to 63% B in 2 min, and then at 63% B for 5 min. The elution was monitored measuring UV absorbance at 214 nm. A total of 96 fractions of 100 µL each were collected and concatenated into 8 fractions/sample.

**LC-MS/MS analysis.** The fractions were analyzed using an Orbitrap Q Exactive HF mass spectrometer equipped with an EASY Spray Source and connected to an UltiMate 3000 RSLC nanoUPLC system (Thermo Scientific). Injected samples were preconcentrated and desalted online using a PepMap C18 nano trap column (length, 2 cm; inner diameter, 75 µm; particle size, 3 µm; pore size, 100 Å; Thermo Scientific) with a flow rate of 3 µL/min for 5 min. Peptide separation was performed using an EASY-Spray C18 reversed-phase nano LC column (Acclaim PepMap RSLC; length, 50 cm; inner diameter, 2 µm; particle size, 2 µm; pore size, 100 Å; Thermo Scientific) at 55 °C and a flow rate of 300 nL/min. The solvent system consisted of 0.1% formic acid, 2% acetonitrile (solvent A) and 98% acetonitrile, 0.1% formic acid (solvent B). Peptides were eluted with a gradient of 4–27% B in 100 min, followed by a 10 min wash (95% B) and 10 min re-equilibration (4% B), step prior the next run. Mass spectra were acquired in a mass-to-charge (m/z) range of 375–1500 with a resolution of 120,000. Automatic gain control target was set to standard mode and maximum injection time to custom mode. Most abundant peptide ions were selected for higher-energy collision dissociation (HCD) with normalized collision energy value set at 35. MS/MS spectra were acquired at a resolution of 50,000, with a maximum injection time of 100 ms and an isolation window of m/z 1.2. The instrument was operated in the positive ion mode for data-dependent acquisition of MS/MS spectra with a dynamic exclusion time of previously selected precursor ions of 45 s.

**Protein identification and quantitative data analysis.** Protein identification and quantification were performed using MaxQuant software[58]. The Uniprot Mus musculus database was used as proteome reference for matching MS/MS spectra. TMT16 quantification of peptide and protein abundances was selected. Cysteine carbamido-methylation was used as a fixed modification; methionine oxidation, N-terminal acetylation as well as arginine, and asparagine deamidation were used as variable modifications for both identification and quantification. Trypsin/P was selected as enzyme specificity with maximum of two missed cleavages allowed. 1% false discovery rate was used as a filter at both protein and peptide levels. After removing contaminants, only proteins with at least two peptides were included in the final data set.

Protein abundances were normalized to the summed abundance of proteins quantified in all fractions for respective sample. This allows better comparison of the proteins in the bound vs unbound samples, since the protein content in the bound samples was less complex. Following the normalization step, protein abundance was further normalized by the corresponding abundance in the linker sample which was used to correct for the instrumental drift between the analyses of the two TMT sets.

**Flow cytometry**
Single-cell suspensions were prepared from mouse PB, SF, or BM samples. For peripheral PBMCs preparation, 30 µL of venous blood were collected by cheek bleeding and buffered in PBS containing heparin on ice. Red blood cells (RBCs) were lysed using 1 mL of ACK buffer twice[59]. SF samples were taken from bilateral knees after termination and buffered in PBS, and BM samples were taken by flushing BM cavity of left femur with PBS. Cells from both SF and BM were filtered using 70-µm cell strainers (BD Biosciences) to remove tissues.

Single cells were pre-incubated with Live/Dead fixable Near-IR dye (ThermoFisher Scientific) plus Fc-block or fluorescent antibody against FCGRs (anti-CD16/32 (2.4G2); PE/Cy7-anti-CD16/32 (93, BioLegend); PE/Cy7-anti-CD16 (S17014E, BioLegend); FITC-anti-CD16 (S17014E, BioLegend), and/or APC-anti-CD16.2 (9E9, BioLegend)) for 20 min at ambient temperature. Next, cells were stained with 50 µL of master mix containing all staining antibodies for 30 min at ambient temperature. The antibodies used in this study include Pacific blue-anti-CD45R (RA3-6B2, BD Biosciences), PerCP/Cy5.5-anti-CD3ε (500A2, BioLegend), Pacific blue-anti-CD11b (M1/70, BioLegend), FITC-anti-CD11b (M1/70, BD Biosciences), BV605-anti-Ly-6C (HK1.4, BioLegend), PE-anti-Ly-6G (1A8, BioLegend), FITC-anti-Ly-6G (1A8, BioLegend), FITC-anti-NK1.1 (PK136, BioLegend), FITC-anti-CD18 (M18/2, BioLegend), APC-anti-CXCR2 (SA044G4, BioLegend), PerCP/Cy5.5-anti-CXCR4 (L276F12, BioLegend), PE/Cy7-anti-C5aR (20/70, BioLegend), PE-anti-Igλ (RML-42, BioLegend), APC-anti-phosphotyrosine (PY20, BioLegend), PE anti-PLCγ2 Phospho (Tyr759) (QA20A56, BioLegend). After the staining, cells were washed and analyzed using Attune NxT flow cytometer. For intra-cellular ROS burst measurement, cells were incubated with Dihydrorhodamine (DHR) 123 (Invitrogen, Thermo-Fisher Scientific) ROS indicator and the fluorescent intensity was then measured by the same flow cytometer under the 530/30 filter excited by a 488-nm laser. M2139, R69-18, or E0320E7 (anti-human IL-34, IgG2b, λ, BioLegend) was used as isotype controls for R69-4 in different scenarios. Intracellular staining was performed typically after surface marker staining. Briefly, cells were fixed by 100 µL of Cytofix/CytoPerm solution (BD Biosciences) for 30 min, and the intracellular targets were stained with corresponding antibodies diluted in Perm/wash buffer (BD Biosciences) for 1 h. After the staining, cells were washed 3 times using the same Perm/wash buffer prior to flow cytometry analysis. All staining procedures were protected from light.

Flow cytometry was performed using Attune NxT flow cytometer, and the data were analyzed using FlowJo software (BD Biosciences). The representative gating strategy for neutrophils was displayed in Fig. S7. Proportion, absolute number, and median fluorescent intensity (MFI) were used to describe gated populations.

## Cytometric bead array (CBA)

Cytokines including IL-1β and CXCL2 were measured by cytometric bead array (CBA) based on Attune NxT flow cytometer platform. The CBA kits for IL-1β and CXCL2 were ordered from BD Biosciences (#562278, detection limit: 0.274 pg/mL) and ThermoFisher Scientific (#EPX01A-26032-901, detection limit: 0.7 pg/mL), respectively. Briefly, serum samples (1:4) or synovial fluids (1:20) were incubated with capture beads coupled with corresponding primary antibody at ambient temperature on a VWR microplate shaker (600 rpm, 1 h), and next the beads were incubated with secondary antibody conjugated with biotin or PE under the same condition. For samples that use biotin conjugated secondary antibody, an additional half an hour of incubation with streptavidin-PE (Invitrogen, ThermoFisher Scientific) was applied. After the incubation, beads' PE fluorescent intensity was measured by Attune NxT flow cytometer.

Relative serum C1q levels were measured by modified CBA using the CXCL2 kit, as C1q binds to IgG ICs and initiates classical pathway of complement activation. Briefly, capturing beads bearing anti-CXCL2 antibody were saturated by excess CXCL2 protein. The beads coupled with CXCL2 ICs were then incubated with diluted serum (1:3) or synovial fluid (1:20) for 1 h under shaking, and then stained using FITC-anti-C1q antibody (JL-1, Nordic Biosite) for 30 min under shaking. FITC fluorescent intensity of the beads was finally measured by the Attune NxT flow cytometer.

Bead-based flow immunoassay was utilized to determine the binding of R69-4 to various peptides. The procedure was detailed elsewhere[60].

## Histology

Mouse hind paws were collected after sacrifice and fixed in 4% paraformaldehyde (PFA) for 10 days with toenails removed. After fixation, paws were washed by tap water and decalcified by the decalcification solution made in-house (10% ethylenediaminetetraacetic acid (EDTA), 2% KOH, 1.2% Tris, and saturated polyvinylpyrrolidone (PVP, 7.5%), pH 6.95). The decalcification buffer was refreshed every 3 days within a period of 4 weeks. After the decalcification procedure, paws were kept in 70% ethanol. Subsequently, paws were gradually dehydrated and embedded in paraffin. Embedded paws were cut into 6-μm sections that were adhered onto SuperFrost Plus slides (Epredia, Thermo Fisher Scientific) which were air-dried thereafter.

Prior to proteoglycan staining, sections were dewaxed in xylene (Sigma-Aldrich) twice, and gradually hydrated and stained with 1% of toluidine blue (T3260, Sigma-Aldrich) for 1 min sharply, followed by de-staining using ddH$_2$O and tap water depending on the microscopic checks with 1-min intervals. Sections were finally visualized under an optical microscope after being covered by Permount mounting medium (Fisher Scientific) and coverslips (Epredia, Fisher Scientific).

In terms of COL2 staining, slides were baked at 60 °C for 30 min prior to hydration, and pepsin (P7000, Sigma-Aldrich) digestion was applied for antigen retrieval (1 g pepsin, 250 ml 0.01 N HCl, 37 °C, 10 min) before staining. After hydration, slides were pre-incubated with R69-4 (IgG λ, 20 μg/mL) or isotype control (E0320E7, IgG λ, Biolegend) for 1 h at 37 °C, and then stained using primary antibodies (Cab4, IgG κ, 20 μg/mL) for 1 h at 37 °C. Finally, FITC-conjugated goat anti-mouse κ antibody (1050-02, Southern Biotech, 1:100, 37 °C, 1 h) was used to stain the bound IgGs with κ light chain. After adequate wash, sections were visualized under a Zeiss LSM800 confocal microscope.

Modified Masson's Trichrome staining kit (#KSC-TRM-1, Nordic Biosite) was employed to stain collagen fibers (all subtypes), and the staining procedures were carried out according to the manufacturer's instructions.

The binding of scFv clones to cartilage was determined by immunohistochemistry (IHC) using frozen neonatal joint tissue. To increase the detection sensitivity, biotinylated anti-His antibody (5 μg/mL) was mixed with scFv in 1:2 molar ratio and incubated at ambient temperature for 1 h. The mix was then added to the tissue sections and incubated at 37 °C for 1 h followed by extra Avidin peroxidase at ambient temperature for 40 min. The IHC staining on adult joints was carried out using the same preparation protocol as COL2 immuno-fluorescence staining, with the difference of using anti-mouse IgG HRP secondary antibody (1031-05, Southern Biotech). The DAB kit (Vector Laboratories) was used to visualize the staining under an optical microscope.

## Confocal imaging

Paraffin embedded sections were visualized using FITC channel only by the Zeiss LSM800 confocal microscope. For evaluating phagocytosis, SF samples were initially collected as described above. SF cells were filtered and stained with PE-anti-Ly-6G (1A8, BioLegend) plus Hoechst (33342, Invitrogen, ThermoFisher Scientific) for 30 min at ambient temperature. After the staining, cells were fixed, permeabilized (BD Cytofix/CytoPerm solution, described above), and further stained intracellularly with AF488-conjugated goat anti-mouse IgG2b (polyclonal, ThermoFisher Scientific) in 1X Perm solution for 1 h. After sufficient wash by the same Perm solution, cells were finally visualized under the 100X objective of the Zeiss LSM800 system using AF488, PE, and Hoechst channels, with compensations been optimized.

## Statistical analysis

Statistical analyses were performed using GraphPad Prism (v9.5.0, GraphPad Software) or Microsoft Excel (v2209, Microsoft Cooperation). The Mann–Whitney $U$ test was used for two groups of quantitative data with small sample size or non-normal distribution. One-way analysis of variance (ANOVA) followed by multiple comparisons with Bonferroni correction was employed for independent measurements with more than 2 groups. For repeated measurements such as arthritis scores, two-way ANOVA followed by post hoc testing with Bonferroni or Dunnett correction was employed to determine the significances between groups. For proteomics data, a two-tailed Student's $t$ test was initially performed, followed by the calculation of false discovery rate (FDR) from the $p$-values and the number of tested variables using the Storey Tibshirani method for multiple hypothesis correction. We considered a two-sided $p < 0.05$ as statistically significant.

## Reporting summary

Further information on research design is available in the Nature Portfolio Reporting Summary linked to this article.

# Data availability

The data supporting the findings of this study are available within the paper (Supplementary Information). The mass spectrometry proteomics data have been deposited to the ProteomeXchange Consortium via the PRIDE partner repository with the dataset identifier PXD044700. All other data and the unique reagents used in this study are available from the corresponding author upon reasonable request. Source data are provided with this paper.

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

## Acknowledgements

This work was supported by grants from the Knut and Alice Wallenberg Foundation, the Swedish Association against Rheumatism, the Swedish Medical Research Council, the Swedish Foundation for Strategic Research, the National Institutes of Health, and the China Scholarship Council. We would like to thank all KM-A technicians, especially Carlos Palestro and Veronika Jansson for taking care of the animals. We acknowledge SciLife Lab at Karolinska Institute for the screening and validation of the recombinant antibodies.

## Author contributions

Z.X., B.X., and R.H. designed the study. Z.X., A.M.G., and D.Z. performed most of the animal experiments and the in vitro experiments, analyzed the data, and interpreted the results. S.L.L. performed mass spectrometry and analyzed the corresponding data. B.X., E.L., L.C., B.L., and D.T. conducted the screening and validation of the recombinant antibodies. Q.L. performed IHC staining. C.G. supervised epitope mapping. A.K. produced the recombinant isotype control. R.S. provided the triple-helical peptides. R.H., M.M., A.M.B., G.B.F., R.A.Z. supervised the experiments. Z.X. and R.H. drafted the first version of the manuscript. All authors contributed to the revision and approved the final version of the submission.

## Funding

## Competing interests

R.H. is the founder of Vacara AB. Z.X. and E.L. are partially employees of Vacara AB. Other authors declare no competing interest.
