## [Peer Review File · Nature Communications]

REVIEWER COMMENTS

Reviewer #1 (expert in rheumatoid arthritis):

In this manuscript Rikard Holmdahl et al. describe their findings regarding a subset of anti-collagen type II antibodies that reduce inflammation via FcγRIII binding. A major strength of the manuscript is the number of different model systems employed as well as utilizing pharmacologic and alternative approaches. The investigators selected clone R69-4 as it bound to collagen and reduced collagen antibody induced arthritis. They identified the binding region and optimized the IgG isotype to IgG2b and hIgG2.

Using a strategic series of genetically engineered mice they demonstrated with BQ.FCGRB2- mice that FcγR2 is not quintessential to the mechanism, but FcγR3 is needed. An additional strength is that the selected antibody does not work in all models of arthritis hence the lack of efficacy in the hGPI325-339 model suggests that there is no cross over in the T cell compartment in modulating the arthritis with this antibody. Additional data addressing the mechanism demonstrates that immune complexes of R69-4 inhibit FcγR3 binding by neutrophils resulting in a reduced signaling cascade that leads to decrease IL-1 and oxidative burst.

Overall the manuscript is well written and logically laid out. This topic is of high interest in the field given the expanded use of biologics in the clinical setting. The methodology is outlined in sufficient detail including in the supplemental files.

A few notes that would improve the manuscript are noted below:

The statistical approaches should be reviewed with a statistician. Many of the differences are quite large and will remain significant, but the use of Student t test applied to small sample numbers instead of a nonparametric test and used as a point to point comparison in assays with repeated measures over time should be re-examined and the main effects reported. Examples-Figure 1c does not seem to have a main effect by area under the curve as demonstrated at the far right of the curves in 1A and 1B, so point to point comparisons are not valid in 1C without a main effect for repeated measures over time. Figures 3 F, 4F, 5A, 6F have repeated measures with time so two way analyses should be considered and the main effect reported with post hoc testing if appropriate rather than selected point to point comparisons.

In the results for Figure 1 could the authors provide an explanation for the delay in the onset of CIA in the R69-4 mice which eventually catch up to the untreated mice? Why is the effect at the point of presumed adaptive immunity rather than later when innate immunity presumably would be the larger component?

The experiment alluded to in the discussion using FcγR3-/- mice and 24G2 treatment in vivo in line 302 of the discussion should be included in the main figures of the paper.

The authors indicate that FcγR3 is necessary for an antibody induced arthritis, but the activity of FcγRIV should be noted as well. FcγRIV is reported to contribute especially to IgG2b-mediated inflammatory responses (doi.org/10.1002/eji.200939884) and is sufficient to induce antibody induced arthritis in the absence of FcγRIII ([DOI:10.4049/jimmunol.1003642](https://doi.org/10.4049/jimmunol.1003642)). In addition results from others using Catchup mice demonstrate that FcγRIV on neutrophils also regulate neutrophil recruitment. There are data buried in supplemental file showing a drop in FcγRIV commensurate with FcγRIII therefore the conclusions should be tempered appropriately. The title also implies that FcγRIII is exclusive in the mechanism rather than a major contributor.

In addition the potential for FcRn blockade is not addressed or discussed as a potential contributing mechanism for injecting an antibody that reduces pathology from antibody mediated disease (*J Immunol.* 2011 Jul 15; 187(2): 1015–1022).

Line 170-The expression of CD18 did not change which would also be true for a mechanism where neutrophils adhere to (marginate) the walls of vessels, reducing the circulating population. Hence the conclusion that the neutrophils are not expanding should be modified. For example, murine treatments that increase type I interferon in mice will also increase the interaction with LFA1 and will result in reduction of circulating neutrophils and influx into the joint of an antibody mediated arthritis.

The conclusions should be mitigated. The antibody described in this manuscript was only effective in antibody mediated arthritis that circumvented the adaptive immune system. This may not be the case in human rheumatoid arthritis as anti-CD20 monoclonal treatment has not been universally effective nor has it put all patients in sustained remission. Hence this agent may have a place in clinical therapy but it is unlikely be a universally remittive agent.

Reviewer #2 (expert in rheumatoid arthritis, complement and antibodies in arthritis):

The manuscript by Zhongwei Xu et al., titled "A subset of antibodies to collagen protects against arthritis by targeting FCGR3 on neutrophils" is an interesting research work by team of highly experienced and well-reputed researchers in the area of rheumatology and complement-mediated diseases. In current study, the authors have shown that a recombinant antibody R69-4 (ScFv) abrogated cartilage antibody-induced arthritis (CAIA) in mice, which is mainly dependent on the complement system. This antibody also protected mice from collagen-induced arthritis (CIA), which is mainly dependent on the T and B cells. Interestingly R69-4 antibody did not protect mice from various other autoimmune diseases such as psoriatic arthritis independent of arthritogenic autoantibodies. The authors have also shown that R69-4 also binds to the complement component, C1q in the synovial fluid. The authors have also provided a detailed mechanism of how the immune complexes (ICs) of R69-4-C1q bind and inhibited signals through FcγR3 receptor. As we know, FcγRs in general plays a very important role in the pathogenesis of rheumatoid arthritis (RA) not only in mice but also in humans. IVIg targeting FcγR3 is being infused in RA patients and patients undergo remission. The decrease in IL-1b secretion by R69-4-C1q ICs confirms the protective effect of this antibody and provide a new therapeutic tool to treat RA for it is well accepted fact since 1939 that RA is not only an IC-mediated disease but also it dependent on the neutrophils and macrophages. Therefore, there is a direct link between R69-4-C1q ICs and FcγR3 receptors. In this manuscript, the authors have provided

a detailed methodology in the supplement section regarding the generation of R69-4 and other recombinant antibodies. The authors have used an appropriate mouse model of arthritis along with mouse models of other autoimmune diseases such as human glucose-6-phosphatase isomerase peptide induced model, mannan-induced psoriasis to confirm all data which is impressive, and this reviewer commends it. The authors have used all appropriate statistical methods based on the nature of data which is acceptable. All figure legends are correct match with the text. Overall, this reviewer agrees with the conclusions and, also inferences made by the authors based on the solid data provided in this manuscript that humanized R69-4 could decrease TNF- α in RA patients. Specifically, when authors have shown that R69-4 does not interfere with the degradation of the pathogenic antibody binding which is very important thus they validated R69-4-C1q-Fc γ R3-IL-1 β mechanism. I have the following minor suggestions before importing this manuscript for publication.

1. Since the major focus of this manuscript is on Fc γ R3 therefore the authors should provide couple of lines in the introduction regarding the role of Fc γ Rs in the clearance of immune complexes. Similarly, please mention in one sentence that C1q is a component of the classical pathway of the complement system for general audience.
2. The authors should label the parts of the knee joint such as synovium, cartilage and bone shown in the graphic abstract or the putative model.
3. Please explain in methods why two different doses of the LPS i.e., 25ug/mouse (line 412) and 50ug/mouse were used to recycle the development of arthritis in different experiments.
4. I think the authors should also highlight the important point in the discussion that the protective effect of R69-4 was not blunted even in the absence of C5 which is important for there is a direct connection or a cross talk between C5a-C5aR1-Fc γ R3 axis. The protective effect of R69-4 is more up-stream to the C5a-C5aR1-C5aR2 axis the level of phagocytosis.
5. Optional comment: If the authors have measured the absolute levels of C1q in the synovial fluid using ELISA or any other method then they can mention it to make sure it is present in the synovial fluid.

Reviewer #3 (expert in Fab and scFv development and phage display):

In this manuscript, the authors describe novel subsets of antibodies against F4 region of COL2 protein which potential for arthritis treatment. This is an important initiative to develop novel drugs for Rheumatoid Arthritis (RA).

Some questions

1-The number of amino acid positions 926-936 in F4 region of COL2 is relative to which protein? Mouse or human? The collagen gene has more than one predicts RNA. It is important to describe the protein ID from NCBI or another database. It was probably described in reference 7, but it is important in this manuscript.

2- Figure 3c and d. What was the animal's age?

3- Lines 319-325: This is a highly speculative claim. Authors must provide data, even bioinformatics such as docking with FCGR2A and R69-4.

4- Lines 336-338 - This is my main question. The authors should provide linear (amino acid) and 3D (conformational) alignment data mainly between COL2 (F4 region) and FCGR3 proteins to map the epitopes of these cross-reacting proteins. It is very likely that they are conformational epitopes.

5- In the IP experiments a lot of proteins could be binding to beads, plastic, and others. I did not identify any control or validation of some protein as real ligand of R69-4 after proteomic.

6- Lines 337-338: I agree the R69-4 could be a novel drug for RA. Which is the limitation? If all RA patients had the same molecular/immunological event it will be putative the best medication. However, the authors tested different murine models of arthritis and found different profiles. So, it is important to appoint the main human RA target and limitations of this candidate novel drug.

RESPONSE TO REVIEWERS' COMMENTS

Reviewer #1 (expert in rheumatoid arthritis):

In this manuscript Rikard Holmdahl et al. describe their findings regarding a subset of anti-collagen type II antibodies that reduce inflammation via FcγRIII binding. A major strength of the manuscript is the number of different model systems employed as well as utilizing pharmacologic and alternative approaches. The investigators selected clone R69-4 as it bound to collagen and reduced collagen antibody induced arthritis. They identified the binding region and optimized the IgG isotype to IgG2b and hIgG2.

Using a strategic series of genetically engineered mice they demonstrated with BQ.FCGRB2- mice that FcγR2 is not quintessential to the mechanism, but FcγR3 is needed. An additional strength is that the selected antibody does not work in all models of arthritis hence the lack of efficacy in the hGPI325-339 model suggests that there is no cross over in the T cell compartment in modulating the arthritis with this antibody. Additional data addressing the mechanism demonstrates that immune complexes of R69-4 inhibit FcγR3 binding by neutrophils resulting in a reduced signaling cascade that leads to decrease IL-1 and oxidative burst.

Overall the manuscript is well written and logically laid out. This topic is of high interest in the field given the expanded use of biologics in the clinical setting. The methodology is outlined in sufficient detail including in the supplemental files.

A few notes that would improve the manuscript are noted below:

The statistical approaches should be reviewed with a statistician. Many of the differences are quite large and will remain significant, but the use of Student t test applied to small sample numbers instead of a nonparametric test and used as a point to point comparison in assays with repeated measures over time should be re-examined and the main effects reported. Examples-Figure 1c does not seem to have a main effect by area under the curve as demonstrated at the far right of the curves in 1A and 1B, so point to point comparisons are not valid in 1C without a main effect for repeated measures over time. Figures 3 F, 4F, 5A, 6F have repeated measures with time so two way analyses should be considered and the main effect reported with post hoc testing if appropriate rather than selected point to point comparisons.

Response: We appreciate the statistical concern raised by the reviewer. We have had our statistical analyses reviewed by two experts in statistics. It was then confirmed that the previous statistical methods employed were indeed not optimal, as suggested by the reviewer. Consequently, we conducted a thorough re-analysis of all the data, by applying non-parametric tests for data with small sample sizes, and two-way ANOVA followed by post hoc testing for repeated measurements, such as arthritis scores. We have now updated the manuscript to reflect these revised statistical analyses accordingly. The significance levels throughout all the analyses remained unchanged, thereby leaving the interpretations unaffected. Relevant descriptions of statistics have been updated in the revised manuscript (Line 679-683).

In the results for Figure 1 could the authors provide an explanation for the delay in the onset of CIA in the R69-4 mice which eventually catch up to the untreated mice? Why is the effect at the point of presumed adaptive immunity rather than later when innate immunity presumably would be the larger component?

Response: One possible explanation is that the initial phase of CIA is more dependent on an antibody attack in the joints whereas the subsequent development is most likely due to a cellular immune response interconnected with a healing response. The R69-4 antibody is targeting the antibody phase as is discussed in the manuscript. Another alternative, contributing, explanation could be the half-life of the injected R69-4 antibodies. Monoclonal mouse IgG2b antibodies, such as R69-4, typically have a half-life of 4-6 days (Vieira P, et al. 1988). In this CIA experiment, the mice received one single dose of R69-4 (mIgG2b) after boost, which did not allow for a long-term remission. We added green arrows to these figures to indicate the day of R69-4 injection. However, it is interesting to note that CAIA mice were "cured" by one dose of R69-4, which is likely attributable to the distinct pathogenesis of CAIA compared to CIA. In CIA, arthritogenic antibodies are continuously produced by plasma cells, which would

promote inflammation when the titer of R69-4 was insufficient to provide protection. Conversely, in CAIA, injected pathogenic antibodies are degraded at a similar rate to R69-4. The principle of antibody half-life applies to these pathogenic antibodies as well.

The experiment alluded to in the discussion using FcγR3^{-/-} mice and 24G2 treatment in vivo in line 302 of the discussion should be included in the main figures of the paper.

Response: We made a deliberate decision not to include the results of these two experiments in the present manuscript, as similar data have been included in another manuscript, where we extensively characterized FCGRs in CAIA development (Related manuscript 1). Using the identical protocol for CAIA modeling described in the present manuscript, we demonstrated the essential and sufficient role of FCGR3 in CAIA by experiments including the above two mentioned by the reviewer (Fig. R1 and Fig. R2). Given that we are unable to include the aforementioned results in the present manuscript, we have rephrased these statements (Line 319-323) in the Discussion section, to be solely based on published evidence.

The authors indicate that FcγR3 is necessary for an antibody induced arthritis, but the activity of FcγRIV should be noted as well. FcγRIV is reported to contribute especially to IgG2b-mediated inflammatory responses (doi.org/10.1002/eji.200939884) and is sufficient to induce antibody induced arthritis in the absence of FcγRIII (DOI:10.4049/jimmunol.1003642). In addition results from others using Catchup mice demonstrate that FcγRIV on neutrophils also regulate neutrophil recruitment. There are data buried in supplemental file showing a drop in FcγRIV commensurate with FcγRIII therefore the conclusions should be tempered appropriately. The title also implies that FcγRIII is exclusive in the mechanism rather than a major contributor.

Response: We highly appreciate the reviewer's suggestion and have included FCGR4 in the Discussion section of the revised manuscript (Line 328-335). Nevertheless, the key question is whether FCGR4 is essential for R69-4 to provide protection. For the reasons summarized below we do not think it is necessary.

1. FCGR4 does play an important role in the K/BxN serum transfer model, but the conclusion drawn from the mentioned paper (Mancardi D, et al. 2011), which suggests that FCGR4 is sufficient in antibody induced arthritis, is probably debatable. The observed effect of FCGR4 blockade in that paper (Fig. 1B) demonstrates its importance, but proving sufficiency would require an experiment where only FCGR4 is present while all other FCGRs are deleted or blocked. Additionally, it is worth noting that the predominant IgG isotype in K/BxN mouse serum is IgG1 (70%-100%) (Maccioni M, et al. 2002), which does not interact with FCGR4 (Dekkers G, et al. 2017) (validated in Fig. R3). FCGR4 is reported to contribute especially to IgG2b-mediated inflammatory responses (Syed S, et al. 2009), as the reviewer mentioned.

2. The CAIA model shares many similarities with K/BxN model, but the two models may have different patterns in terms of FCGR activation. In the manuscript mentioned before (Related manuscript 1), we demonstrate that FCGR3 is essential and sufficient for CAIA development (Fig. R1 and Fig. R2). Additionally, we class-switched R69-4 to mIgG1 isotype that does not interact with FCGR4 (Fig. R3) and administered it in both Cab4 and mIgG1 cocktail induced arthritis model. We observed a potent protective effect of R69-4-IgG1 against Cab4 induced arthritis (Fig. S1b), as well as IgG1 cocktail induced arthritis (Fig. R4), which excludes the necessity of R69-4 to ligate to FCGR4 for ameliorating inflammation.

Although FCGR4 is not necessary for CAIA or R69-4 as we demonstrated, it contributes to arthritis development to some extent. The substantial downregulation of both FCGR3 and FCGR4 upon R69-4 (IgG2b) injection suggests that it may also abrogate K/BxN arthritis. According to the reviewer's suggestions, we have tempered our interpretations to avoid the misconception of an exclusive mechanism. We have also revised the title to underscore a major effect. We sincerely appreciate this comment that helps ensure the objectivity and fairness of our conclusions.

In addition the potential for FcRn blockade is not addressed or discussed as a potential contributing mechanism for injecting an antibody that reduces pathology from antibody mediated disease (J Immunol. 2011 Jul 15; 187(2): 1015–1022).

Response: We acknowledge the reviewer for highlighting this important issue. We excluded FcRn as a potential mechanism based on the observation that R69-4 did not shorten the half-life of the injected pathogenic antibodies in the CAIA model (Fig. 2a). Considering that FcRn is responsible for antibody recycling and protects monomeric pathogenic antibodies from degradation, it is unlikely that FcRn was blocked after R69-4 injection. Blocking FcRn would have a significant impact on the recycling of the injected pathogenic antibodies, and would result in a substantial change in the pharmacokinetic profile of these antibodies (Blumberg L, et al. 2019, Nixon A, et al. 2015). Accordingly, we have included the rationale in the manuscript regarding the exclusion of FcRn (Line 117-119).

Line 170-The expression of CD18 did not change which would also be true for a mechanism where neutrophils adhere to (marginate) the walls of vessels, reducing the circulating population. Hence the conclusion that the neutrophils are not expanding should be modified. For example, murine treatments that increase type I interferon in mice will also increase the interaction with LFA1 and will result in reduction of circulating neutrophils and influx into the joint of an antibody mediated arthritis.

Response: We concur with the notion that such a scenario can indeed occur, but it is not likely in the case of R69-4. In CAIA, the reduction of circulating neutrophils due to their tethering to blood vessels is transient, followed by neutrophil extravasation cascade driven by chemokines or other mediators. Here, we did not speculate that R69-4 could increase neutrophil adherence to the walls of blood vessels which further contributes to the observed remarkable decrease of circulating neutrophils, because of the following reasons:

1. We observed a long-term suppression of neutrophil expansion (Fig. 3b). It is not likely that neutrophils are permanently immobilized on vascular endothelium by R69-4 without transmigrating across the blood vessels or homing back to the blood stream.

2. After R69-4 injection, we observed a dramatic decrease in circulating neutrophils (Fig. 3c), along with an increase of neutrophils in bone marrow (Fig. S3a). This observation is more likely due to the decreased demand for recruiting neutrophils rather than enhanced adherence to blood vessels.

3. Unchanged CD18 expression after R69-4 treatment probably suggests unchanged capacity of neutrophils to adhere to the walls of blood vessels, rather than increased interactions.

4. We observed a recovery of circulating neutrophils after IL-1 β administration in mice treated with R69-4 (Fig. 3g). If only R69-4 reduces circulating neutrophils by enhanced tethering, no increase should be observed after IL-1 β administration, because these expanded neutrophils would also be captured by vascular membranes.

Therefore, we believe that the conclusion that neutrophil expansion is suppressed after R69-4 injection in CAIA remains solid.

The conclusions should be mitigated. The antibody described in this manuscript was only effective in antibody mediated arthritis that circumvented the adaptive immune system. This may not be the case in human rheumatoid arthritis as anti-CD20 monoclonal treatment has not been universally effective nor has it put all patients in sustained remission. Hence this agent may have a place in clinical therapy but it is unlikely be a universally remittive agent.

Response: We acknowledge the reviewer's perspective that the original conclusions were overly optimistic, and we have mitigated our statements accordingly. At this stage, extrapolating its potent efficiency from mouse studies to RA patients, particularly in the established chronic phase, is probably premature. We totally agree that it cannot become a universally effective medication for all types of RA, and we have discussed about the possible applications in certain subtypes in the revised manuscript (Line 368-375).

Overall, we would like to express our gratitude to the reviewer for the diligent scrutiny, which has been instrumental in helping us identify and correct mistakes and avoid over-interpretation. These insightful comments particularly regarding neutrophil biology and FCGR functions, have significantly contributed to the improvement of our revised manuscript.

Reviewer #2 (expert in rheumatoid arthritis, complement and antibodies in arthritis):

The manuscript by Zhongwei Xu et al., titled “A subset of antibodies to collagen protects against arthritis by targeting FCGR3 on neutrophils” is an interesting research work by team of highly experienced and well-reputed researchers in the area of rheumatology and complement-mediated diseases. In current study, the authors have shown that a recombinant antibody R69-4 (ScFv) abrogated cartilage antibody-induced arthritis (CAIA) in mice, which is mainly dependent on the complement system. This antibody also protected mice from collagen-induced arthritis (CIA), which is mainly dependent on the T and B cells. Interestingly R69-4 antibody did not protect mice from various other autoimmune diseases such as psoriatic arthritis independent of arthritogenic autoantibodies. The authors have also shown that R69-4 also binds to the complement component, C1q in the synovial fluid. The authors have also provided a detailed mechanism of how the immune complexes (ICs) of R69-4-C1q bind and inhibited signals through FcγR3 receptor. As we know, FcγRs in general plays a very important role in the pathogenesis of rheumatoid arthritis (RA) not only in mice but also in humans. IVIg targeting FcγR3 is being infused in RA patients and patients undergo remission. The decrease in IL-1b secretion by R69-4-C1q ICs confirms the protective effect of this antibody and provide a new therapeutic tool to treat RA for it is well accepted fact since 1939 that RA is not only an IC-mediated disease but also it dependent on the neutrophils and macrophages. Therefore, there is a direct link between R69-4-C1q ICs and FcγR3 receptors. In this manuscript, the authors have provided a detailed methodology in the supplement section regarding the generation of R69-4 and other recombinant antibodies. The authors have used an appropriate mouse model of arthritis along with mouse models of other autoimmune diseases such as human glucose-6-phosphate isomerase peptide induced model, mannan-induced psoriasis to confirm all data which is impressive, and this reviewer commend it. The authors have used all appropriate statistical methods based on the nature of data which is acceptable. All figure legends are correct match with the text. Overall, this reviewer agrees with the conclusions and, also inferences made by the authors based on the solid data provided in this manuscript that humanized R69-4 could decrease TNF-α in RA patients. Specifically, when authors have shown that R69-4 does not interfere with the degradation of the pathogenic antibody binding which is very important thus they validated R69-4-C1q-FcγR3-IL-1β mechanism. I have the following minor suggestions before importing this manuscript for publication.

1. Since the major focus of this manuscript is on FcγR3 therefore the authors should provide couple of lines in the introduction regarding the role of FcγRs in the clearance of immune complexes. Similarly, please mention in one sentence that C1q is a component of the classical pathway of the complement system for general audience.

Response: We thank the reviewer for this comment. In the revised manuscript, we have added relevant text regarding the clearance of ICs by FCGRs as well as the role of C1q in complement activation (Line 54-60).

2. The authors should label the parts of the knee joint such as synovium, cartilage and bone shown in the graphic abstract or the putative model.

Response: Following the suggestion, we have now revised the graphic abstract as well.

3. Please explain in methods why two different doses of the LPS i.e., 25ug/mouse (line 412) and 50ug/mouse were used to recycle the development of arthritis in different experiments.

Response: At the screening stage of recombinant antibodies, we only used two arthritogenic antibodies (M2139 and CIIC1) for CAIA modeling. This two-antibody cocktail is less arthritogenic than the subsequently developed Cab4 cocktail containing 4 monoclonal antibodies that induces arthritis potently (Li Q, et al. 2022). The usage of a double dose of LPS for M2139+CIIC1 induced arthritis model was an attempt to increase the disease severity, but despite this, the arthritis severity was still relatively low (Fig. S1a). Once we established a standard protocol (Cab4) for inducing CAIA, we fixed the LPS dose at 25 μg for Cia9i mice. It is important to note that LPS was not administrated in any CAIA experiments involving FCGR2B⁻ animals. We have provided the rationale of using two doses of LPS in the supplementary materials (Supplementary brochure, 3.6 CAIA).

4. I think the authors should also highlight the important point in the discussion that the protective effect of R69-4 was not blunted even in the absence of C5 which is important for there is a direct connection or a cross talk between C5a-C5aR1-Fc γ R3 axis. The protective effect of R69-4 is more up-stream to the C5a-C5aR1-C5aR2 axis the level of phagocytosis.

Response: We appreciate this comment which highlights an important aspect that we did not mention previously. Apart from complement activation that finally leads to MAC formation, various components in the complement system (e.g., C5a), play a significant role in arthritis development. Specifically, C5a and C5a receptors have been identified as an important pathway for antibody-mediated inflammation in wild-type animals. We have expanded our discussion on this topic according to the reviewer's suggestion (Line 336-343).

5. Optional comment: If the authors have measured the absolute levels of C1q in the synovial fluid using ELISA or any other method then they can mention it to make sure it is present in the synovial fluid.

Response: We measured the levels of C1q in synovial fluid using modified CBA based on flow cytometry as we did in serum. But quantifying the absolute levels can be tricky using this method. To this end, we showed the relative levels which should be sufficient to ensure that C1q is present in the synovial fluid, as we observed a significant decrease of FITC fluorescent intensity (conjugated with anti-C1q secondary antibody) of beads incubated with CAIA+R69-4 SF compared to CAIA SF, which is consistent with the findings from sera as well as the findings from mass spectrometry. We have included these results in Fig S5e.

Overall, we would like to thank the reviewer for the strong endorsement of our study and all the thoughtful suggestions. The professional perspective in the complement system has contributed to the interpretation of our results.

Reviewer #3 (expert in Fab and scFv development and phage display):

In this manuscript, the authors describe novel subsets of antibodies against F4 region of COL2 protein which potential for arthritis treatment. This is an important initiative to develop novel drugs for Rheumatoid Arthritis (RA).

Some questions

1-The number of amino acid positions 926-936 in F4 region of COL2 is relative to which protein? Mouse or human? The collagen gene has more than one predicts RNA. It is important to describe the protein ID from NCBI or another database. It was probably described in reference 7, but it is important in this manuscript.

Response: We appreciate the reviewer for highlighting this confusion. Our position numbering referred to the sequence of amino acids forming the triple-helical domain, a numbering system often used to describe collagens. To provide a more easily understandable and consistent numbering system, we have applied UniProt IDs as references and made the necessary clarifications in the revised manuscript (Line 50-51). Furthermore, we have opted for the canonical sequences to avoid potential confusion arising from various isoforms (P02458-2 for human and P28481-3 for mouse). The F4 region is highly conserved in both human and mouse, with the same sequence and location in both species.

2- Figure 3c and d. What was the animal's age?

Response: The animals involved in Fig. 3c were aged from 96 to 103 days on the day of blood sampling, and the animals involved in Fig. 3d were aged from 84 to 93 days on the day of SF sampling. We indicated the range of age of all included animals in the Methods-Animal models section (Line 418).

3- Lines 319-325: This is a highly speculative claim. Authors must provide data, even bioinformatics such as docking with FCGR2A and R69-4.

Response: We thank the reviewer for this comment. We made this speculation from mouse to human based on the evolutionary relationship between mouse FCGR3 and human FCGR2A (Lejeune J, et al. 2019). It has been demonstrated that human FCGR2A interacts with all human IgG isotypes with their IC form through Fc-FCGR interaction (Bruhns P, et al. 2009). We should have highlighted that we propose a potential R69-4-Fc-FCGR2A interaction rather than an R69-4-Fab-FCGR2A specific binding. In any case, as the reviewer suggested, it is improper to have highly speculative claims without supporting experimental data. Therefore, we have rephrased these statements in Discussion in a more conservative manner (Line 353-357).

4- Lines 336-338 - This is my main question. The authors should provide linear (amino acid) and 3D (conformational) alignment data mainly between COL2 (F4 region) and FCGR3 proteins to map the epitopes of these cross-reacting proteins. It is very likely that they are conformational epitopes.

Response: We are not sure that we understand the question correctly as there could be two different interpretations. A direct interpretation would be that we need to show the molecular similarity between the R69-4 interaction sites on COL2 and on FCGR3. If this is the question, we need to admit that we had not clearly explained how we think R69-4 operates. To our interpretation R69-4 interacts with various joint proteins, including those with triple helical collagen structures, to form unique immune complexes. These immune complexes interact with FCGR3 on neutrophils, presumably through Fc domains of antibodies within the immune complexes. However, to directly answer the reviewer's question, as we interpreted it, we have conducted an alignment analysis between the F4 epitope and FCGR3. As expected, our analysis did not reveal linear alignment between COL2 (F4 region) and FCGR3 (Fig. R5). Conducting a conformational alignment between triple helical COL2 and other proteins poses a challenge due to the immense size of collagens, which are composed of three intertwined alpha chains with complex assembly and inherent flexibility. Alternatively, we performed conformational alignment between FCGR3 and a short triple-helical peptide containing the F4 epitope (6HG7). The alignment yielded a relatively low level of similarity (17% identity, Fig. R6), with the aligned sequence primarily located in the intracellular domain of FCGR3, which appears to be biologically irrelevant. To directly address the possibility whether R69-4 interacted with FCGR3

through its antigen binding site or through its Fc domains we made an ELISA test. As expected, intact R69-4 exhibited binding to FCGR3, whereas R69-4-Fab did not (Fig. S5d). These results confirm that R69-4 interacts with FCGR3 solely through its Fc portion, rather than via specific binding through its Fab region. We have emphasized this crucial point by changing the corresponding subtitle in the Results section to " R69-4 binds neutrophils through Fc-Fc receptor interaction". We realize that there is confusion within our Graphic Abstract, and we have now corrected this to better reflect how we think R69-4 containing immune complexes interact with FCGR3.

Alternatively, the question by the reviewer could also be interpreted as the molecular mimicry of the triple helical F4 epitope on other proteins. Such triple-helical collagen structure can be found in many proteins in the joints, for example, various collagens, C-type lectin receptors and complement C1q. Since we provide evidence of a direct interaction with the collagen-like part of C1q we now also attempted to align triple helical F4 epitope (6HG7) with C1q. The alpha chain of C1q exhibited a relatively high level of alignment with the alpha chain of 6HG7 (39% identity), as shown in Fig. R7.

In addition, we realize that our inaccurate use of terminology may lead to misunderstandings on the part of the reviewer. We propose that R69-4 protects against arthritis through ligand (R69-4 containing IC) and receptor (FCGR3) interaction (Fig. 5a), rather than through antibody (R69-4) and epitope (FCGR3) binding. We have now changed "ligation" to "Fc-Fc receptor interaction" throughout the revised manuscript to avoid further misunderstandings.

5- In the IP experiments a lot of proteins could be binding to beads, plastic, and others. I did not identify any control or validation of some protein as real ligand of R69-4 after proteomic.

Response: We acknowledge the reviewer's concern regarding potential artificial cross-reactivities in the immunoprecipitation (IP) experiments. To mitigate this effect, we applied isotype control strategy. We excluded candidates that showed significant binding to both R69-4 and the isotype control (blue dots in Fig. 6d). By doing so, we could theoretically eliminate the impact of unspecific binding to beads and plastic materials. Furthermore, during the data analysis process after proteomics, we conducted correlation analysis between R69-4 and the isotype control to somewhat minimize the possibility of false findings (Fig. 6e).

We agree that a validation confirming the proteomic results could strengthen the conclusion. We attempted to do this before submission but found that the majority of the proposed proteins are not commercially available. However, in response to the comment, we have done our utmost effort to search and order all commercially available alternatives, including fusion proteins or human counterparts. With all the ordered proteins detected in the proteomics, we could confirm the binding to R69-4 (Line 280-285, Fig. S6a). To the best of our ability, these results validate the findings from our proteomic analysis. We have also updated Table. S3 with the validation results accordingly.

6- Lines 337-338: I agree the R69-4 could be a novel drug for RA. Which is the limitation? If all RA patients had the same molecular/immunological event it will be putative the best medication. However, the authors tested different murine models of arthritis and found different profiles. So, it is important to appoint the main human RA target and limitations of this candidate novel drug.

Response: We appreciate the reviewer's endorsement of R69-4 as a potential novel drug for RA, as well as the objective perspective regarding its limitations. As we observed a strong effect of R69-4 in prohibiting antibody mediated pathogenesis but not in T cell driven arthritis, its efficacy in chronic RA may be limited. Instead, it could be more effective at an early stage of acute RA, where antibody mediated attack is ongoing. We have elaborated on these concerns in the Discussion section as per the reviewer's suggestion (Line 368-375).

Overall, we extend our gratitude to the reviewer for all the above constructive comments particularly concerning structure biology. These insightful comments have helped us mitigate misconceptions from our manuscript.

Figures for review

Fig. R1. REDACTED

Fig. R2. REDACTED

Fig. R3. Binding between IgG isotypes and FCGR4.

a. Binding of mouse monomeric IgGs to FCGR4: 15A (IgG1), ACC1 (IgG2a), M2139 (IgG2b).

b. Binding of mouse IgG-ICs to FCGR4: 15A (IgG1), ACC1 (IgG2a), M2139 (IgG2b).

Fig. R4. REDACTED

Fig. R5 Linear alignment between FCGR3 and COL2

Fig. R6 Conformational alignment between Col2A1-F4 and FCGR3

Fig. R7 Conformational alignment between Col2A1-F4 and C1qA

REVIEWERS' COMMENTS

Reviewer #1 (expert in rheumatoid arthritis):

No further comments.

Reviewer #2 (expert in rheumatoid arthritis, complement and antibodies in arthritis):

The authors have provided answers in-depth supported by experimental data to all my questions and all relevant changes have been shown in the manuscript.

Reviewer #3 (expert in Fab and scFv development and phage display):

The authors answered all questions clearly. I am satisfied and consider the manuscript ready for publication

RESPONSE TO REVIEWERS' COMMENTS

Reviewer #1 (expert in rheumatoid arthritis):

No further comments.

Reviewer #2 (expert in rheumatoid arthritis, complement and antibodies in arthritis):

The authors have provided answers in-depth supported by experimental data to all my questions and all relevant changes have been shown in the manuscript.

Reviewer #3 (expert in Fab and scFv development and phage display):

The authors answered all questions clearly. I am satisfied and consider the manuscript ready for publication

RESPONSE TO REVIEWERS

Response: We sincerely thank our 3 professional reviewers for the significant time and meticulous effort they dedicated during this process. These constructive comments have been instrumental in helping us improve the quality of our manuscript.